



# Hydrological evaluation of open-access precipitation data using SWAT at multiple temporal and spatial scales

Jianzhuang Pang[1], Huilan Zhang[1*], Quanxi Xu[2], Yujie Wang[1], Yunqi Wang[1], Ouyang Zhang[2], Jiaxin Hao[1]

[1]Three-gorges reservoir area (Chongqing) Forest Ecosystem Research Station, School of Soil and Water Conservation, Beijing Forestry University, Beijing 100083, China

[2] Bureau of Hydrology, Changjiang Water Resources Commission of the Ministry of the Water Resources, Wuhan 430010, China

*Correspondence to*: Huilan Zhang (zhanghl@bjfu.edu.cn)

**Abstract.** Temporal and spatial precipitation information is key to conducting effective hydrological process simulation and forecasting. Herein, we implemented a comprehensive evaluation of three selected precipitation products in the Jiang River Watershed (JRW) located in southwest China. A number of indices were used to statistically analyze the differences between two open-access precipitation products (OPPs), i.e. Climate Hazards Group Infra-Red Precipitation with Station (CHIRPS) and CPC-Global (CPC), and the rain gauge (Gauge). The three products were then categorized into sub-basins to drive SWAT simulations. The results show: (1) the three products are highly consistent in temporal variation on a monthly scale, yet distinct on a daily scale. CHIRPS is characterized by overestimation of light rain, underestimation of heavy rain, and a high probability of false alarm. CPC generally underestimates rainfall of all magnitudes; (2) All three products satisfactorily reproduce the stream discharges at the JRW outlet with better performance than the Gauge model. On a temporal scale, the OPPs are inferior with respect to capturing flood peak, yet superior at describing other hydrograph features, e.g. rising and falling processes and base flow. On a spatial scale, CHIRPS offers the advantage of deriving smooth, distributed precipitation and runoff due to its high resolution; (3) The water balance components derived from SWAT models with equal simulated streamflow discharges are remarkably different between the three precipitation inputs. The precipitation spatial pattern results in an increasing surface flow trend from upstream to downstream. The results of this study demonstrate that evaluating precipitation products using only streamflow simulation accuracy will conceal the dissimilarities between these products. Hydrological models alter hydrologic mechanisms by adjusting calibrated parameters. Specifically, different precipitation detection methods lead to temporal and spatial variation of water balance components, demonstrating the complexity in describing natural hydrologic processes.


## 1. Introduction

Precipitation has been established as the most significant meteorological parameter with respect to forcing and calibrating hydrologic models; as its spatial–temporal variability considerably influences hydrological behavior and water resource availability (Galván et al., 2014; Lobligeois et al., 2014; Roth & Lemann, 2016). Previous studies have demonstrated that reducing precipitation data uncertainty has a sizeable impact on stabilizing model parameterization and calibration (Mileham et al., 2008; Cornelissen et al., 2016; Remesan & Holman, 2015). However, accurate portrayal of authentic basin rainfall inputs' spatial–temporal variability has severe limitations (Bohnenstengel et al., 2011; Liu et al., 2017), and successfully acquiring such data from available resources has typically posed numerous challenges for hydrologic modeling (Long et al., 2016; Zambrano-Bigiarini et al., 2017).

Conventionally, hydrologists have regarded gauge measurements as actual rainfall (Zhu et al., 2015; Musie et al., 2019), and used point rainfall measurements from rain gauges to perform spatial interpolation and illustrate rainfall field in basin/sub-basin regions (Weiberlen & Benitez, 2018; Belete et al., 2019). Ideally, if rain gauges are positioned with reasonable density and uniform distribution, the spatial precipitation variation described by this method is still valid (Duan et al., 2016). However, in remote or developing regions, the meteorological stations are usually scarce and irregularly distributed, resulting in inconsistent and erroneous distributed rainfall field (Hwang et al., 2011; Peleg et al., 2013; Cecinati et al., 2018; Luo et al., 2019). In other cases, when data observation is accidentally missing, the data quality might be unreliable (Alijanian et al., 2017; Sun et al., 2018). These challenges make ground-based precipitation measurements subject to large uncertainties for driving and calculating hydrologic models.

Over the last few decades, open-source precipitation products (OPPs) have provided a promising alternative for detecting temporal and spatial precipitation variability (Qi et al., 2016; Jiang et al., 2018; Jiang et al., 2019). A variety of studies have demonstrated the accuracy differences among the various OPPs, as well as within the same OPP among different regions (Gao et al., 2017 Zambrano-Bigiarini et al., 2017; Lu et al., 2019; Wu Y et al., 2019). Sun et al (2018) evaluated and compared the advantages and disadvantages of 29 OPPs with different spatial and temporal resolutions with respect to their ability to describe global precipitation. Unlike most OPPs with a spatial resolution of 0.25°~ 0.5°, the Climate Hazards Group Infra-Red



Precipitation with Station (CHIRPS), a "satellite-gauge" type precipitation product, provides very fine spatial resolution (Funk et al., 2015), with 0.05° being equivalent to a resolution of one gauge station for every 30.25 km$^2$ area. This characteristic has facilitated widespread use and consistent admiration of CHIRPS in recent years. Duan Z et al. (2019) evaluated three

precipitation products in Ethiopia and found that CHIRPS performed best among them; Lai et al. (2019) used PERSIANN-CDR and CHIRPS driven hydrological simulation in the Beijiang River basin of China, and determined that CHIRPS performed significantly better than the PERSIANN-CDR. The Climate Prediction Center Gauge-Based Analysis of Global Daily Precipitation (CPC-Global) is a unified precipitation analysis product from the National Oceanic and Atmospheric Administration (NOAA) Climate Prediction Center (CPC) (Xie et al., 2007) that contains unified precipitation data collected

from > 30,000 monitoring stations belonging to the WMO Global Telecommunication System, Cooperative Observer Network, and other national meteorological agencies. The product was created using the optimal interpolation objective analysis technique and the data is considered to be relatively accurate. Tian et al. (2010) used CPC as reference data to evaluate the applicability of GSMaP in the United States; and Beck et al. (2017b) used CPC to modify the MSWEP data product they built. Marked differences have been found in the accuracy and spatio-temporal patterns of different precipitation datasets, which

highlights the critical importance of dataset selection for both scientific researchers and decision makers.

The hydrological model or rainfall-runoff model is an important tool for understanding hydrological processes and aids water resources operation decision-makers (Yilmaz et al., 2012; Yan et al., 2016; Wu J et al., 2019). Most frequently used hydrologic models have been shown to efficiently incorporate data from rain gauges, while open-access precipitation has also been continuously improved and adopted into different modules that evaluate its performance in simulating watershed runoff

(Bhuiyan et al., 2019; Solakian et al., 2019). Among all the various existing hydrologic models, the Soil & Water Assessment Tool (SWAT) is widely employed by the scientific community and others interested in watershed hydrology research and management (Price et al., 2013; Wang et al., 2017; Li et al., 2018; Qiu et al., 2019). ~ 4000 peer-reviewed papers in reputable academic journals worldwide (SWAT literature database, from 1984 to 2020) have used SWAT modeling results to support their scientific endeavors. Moreover, a new version (SWAT+) is currently in development that will provide a more flexible

spatial representation of interactions and processes within a watershed (Volk et al., 2009; Gabriel et al., 2014; Ayana et al., 2015; Jin et al., 2018). As mentioned above, numerous researchers concur that designing accurate watershed models requires



realistic depiction of temporal and spatial precipitation variability. Huang et al. (2019) used hourly, sub-daily, and diurnal precipitation data to simulate runoff from Baden-Württemberg state in Germany and found a positive correlation between higher rainfall temporal resolution and model performance. As such, hydrological processes simulated by the SWAT model,

that were based on environmental data lacking accurate regional precipitation distribution figures, will unquestionably be faulty and unreliable. For example, Lobligeois et al. (2014) used rain-gauge measurements (2,500 stations within an area of 550,000 $km^2$), and Weather radar network data with a spatial resolution of 1km, to simulate runoff from France. Their results clearly showed that the higher resolution radar data significantly improved the simulation accuracy.

To date, the effect of combined ground-based and satellite-based precipitation estimates on streamflow simulation

accuracy is not well understood, particularly when the data covers a variety of temporal and spatial resolutions. Hence, this study aims to elucidate these unknowns. More importantly, hydrologic models are expected to describe internal hydrologic processes and subsequently present a unique interpretation of the water balance components (Pellicer-Martínez et al., 2015; Tanner & Hughes, 2015; Wang et al., 2018); yet very limited studies have been conducted to investigate the effects of temporal and spatial resolution on hydrologic processes or water balance components. Thus, this fundamental issue must be addressed

before hydrologic modeling with open-access precipitation datasets can be utilized at maximum capacity; as without a thorough understanding of the water cycle's inner processes, the hydrologic models may be highly misleading and facilitate inappropriate management decisions. Bai & Liu (2018) used an HIMS model to simulate the runoff driven by CHIRPS, CMORPH, PERSIANN-CDR, TMPA 3B42, and MSWEP at the source regions of the Yellow River and Yangtze River basins in the Tibetan Plateau. They reported that parameter calibration significantly counterbalanced the impact of diverse

precipitation inputs on runoff modeling, resulting in substantial differences in evaporation and storage estimates. Their research helps enhance our understanding of how water balance components are impacted by precipitation data and hydrologic model parameters. However, evidence for water balance component variations under the influence of different precipitation inputs have not been fully investigated. In this study, we aimed to verify the ability of CPC-global and CHIRPS to accurately simulate watershed runoff, and analyze how different OPP characteristics modify the hydrological processes simulation.

The Yangtze River is the largest river in China. Its upper segment is home to its primary tributary with the largest drainage area—the Jialing River, which exhibits spatial heterogeneity with respect to climate, geomorphologies, and land cover

conditions. Over the past six decades, anthropogenic activity in conjunction with climate change have substantially reduced

the drainage basin's streamflow (Meng et al., 2019), which significantly impacts the inflow condition of the three Gorge

reservoir. Therefore, it is debatably imperative to elucidate how varying rainfall characteristics impact runoff and hydrological

processes in the Jialing River Watershed (JRW), especially in spatio-temporal dimensions. Given the above considerations,

herein, we attempt to: (1) statistically quantify the differences of ground-based and typical open-source precipitation datasets

in the JRW, (2) evaluate the performances of different precipitation datasets in simulating the watershed streamflow using

SWAT, and (3) investigate the potential behaviors of different precipitation dataset in describing hydrologic processes. All of

the above objectives were analyzed on temporal and spatial scales. The goal of this study was to surpass an accuracy assessment

of rainfall estimates, and evaluate the use of diverse precipitation data types as model operation forcing data and in hydrologic

process portrayal.

## 2. Materials and methods

### 2.1. Study area

The Jialing River is the primary tributary of the Yangtze River, with the larges drainage area of 159,812 km$^2$ and a total

length of ~1345 kilometers. The JRW is situated between 29°17′30′′ N and 34°28′11′′ N and 102°35′36′′ E and 109°01′08′′ E

and geographically extends over the northern part of the transition zone under the eastern Tibet Plateau. The JRW's elevation

difference is ~5000 m and the average gradient is ~ 2.05 ‰. Due to the sharply changing topographic gradients, the area

features northwest highlands, northern mid-low mountains, middle-eastern hills, and southern plains (Fig. 1). The

hydrometeorological conditions follow a similar spatial distribution pattern, i.e., relatively colder and drier in the north and

warmer and wetter in the south (Meng et al., 2019). The long-term annual precipitation, based on records from 1956–2018,

ranges from ~ 900 to 1200 mm, a product of southwest China's warm, low latitude air. The rainfall is mainly concentrated

from May to September, which accounts for about 60% of the annual precipitation. The annual average temperature ranges

from 4.3 to 27.4°C, and the annual average evapotranspiration ranges from 800 to 1000 mm. The daylight duration is ~1890

h/year, with an annual wind speed of 0.7 ~ 1.8 m/s. Annual relative humidity ranges from 57 to 79 % (Herath et al., 2017). The

controlled hydrologic Beibei station is gauged at the JRW outlet, and the long-term mean annual runoff is ~ $6.55 \times 10^8$ m$^3$

year$^{-1}$, according to the Changjiang Sediment Bulletin. The Digital Elevation Model (DEM), river network, and meteorological

stations are shown in Fig.1.

### 2.2. Data sources

The data required for SWAT modeling and validation consisted of geographic information, meteorological, and

hydrological datasets.

### 2.2.1 Geographic information dataset

The Geographic dataset included the DEM, land use and land cover data (LULC), and soil properties. SRTM 90 m

resolution data was the DEM used in this study and was provided by the Geospatial Data Cloud (http://www.gscloud.cn/).

LULC data was obtained by manual visual interpretation based on 2010 Landsat TM/ETM remote sensing images, which were

preprocessed by Beijing Digital View Technology Co., Ltd, with a spatial resolution of 30 m. The data included six primary

classifications—cultivated land, woodland, grassland, water area, construction land, and unused land, as well as 25 secondary

classifications. The cultivated land's classification accuracy was 85 %, and other data classification accuracies reached 75 %.

The soil data included a soil type distribution map and soil attribute database. The soil type distribution map is a product of

the second national land survey provided by the Nanjing Institute of Soil Research, Chinese Academy of Sciences. It depicted

a spatial resolution of 1 km and used the FAO-90 soil classification system. Harmonized World Soil Database v1.2, which was

provided by the Food and Agriculture Organization (FAO), was the soil attribute database used in this study and can be

downloaded from http://www.fao.org/land-water/databases-and-software/hwsd/zh/. Most soil attribute data can be obtained

directly from HWSD v1.2, such as soil organic carbon content, soil profile maximum root depth, and soil concrete gradation,

etc. Parameters that cannot be directly obtained from the HWSD, e.g., texture class, matrix bulk density, field capacity-wilting

point, and saturation hydraulic coefficient, can be calculated from the acquired data using the Soil-Air-Water Field & Pond

Hydrology model developed by Washington State University. All the above geographic information data were processed by

ArcMap 10.2 to obtain 250 m spatial resolution data of the JRW, using the Beijing_1954_GK_Zone_18N Projection coordinate system and GCS_Beijiing_1954 Geographic coordinate system.

### 2.2.2 Meteorological and hydrological dataset

Daily observed discharges are documented from 1997–2018 at Beibei hydrological station. The daily meteorological records—including precipitation, temperature, relative humidity, sunshine hours, and wind velocity, were measured by 20 meteorological stations in and around the JRW, which were provided by the China meteorological data network (http://data.cma.cn/). The solar radiation data required for establishing the meteorological database was calculated using the sunshine hours ($n$), and the calculation method consisted of employing the solar radiation ($Rs$) index in the FAO-56 Penman-

Monteith method.

The most recent gridded format CHIRPS product (V2.0 datasets) was completed and released in February 2015. The dataset spans from 1981 to the present and provides daily precipitation data with a spatial resolution of 0.05° in a pseudo-global coverage of 50° N - 50° S. The data is available for download at http://chg.geog.ucsb.edu/data/chirps/. The CHIRPS product is composed of various types of precipitation products, including ground measurement, remote sensing, and reanalysis

data—e.g. The Climate Hazards Group Precipitation Climatology (CHPclim) is built from monthly precipitation data supplied by the United Nations FAO, Global Historical Climate Network (CHCN), Cold Cloud Duration from NOAA, and TRMM 3B42 Version 7 from NASA, etc. Essentially, the above-mentioned data were synthesized into 5-day rainfall records, and then the rain gauge observations from multiple data sources were used to correct the deviation, which was further interpolated to daily scaled CHIRPS product. More detailed information about CHIRPS products is available in Funk et al. (2015).

CPC-Global, in gridded format, is the first generation product of NOAA's ongoing CPC unified precipitation project. This product offers daily precipitation estimates from 1998 to the present, at a spatial resolution of 0.5° over land. Daily precipitation data for CPC-Global can be downloaded from: http://ftp.cpc.ncep.noaa.gov/precip/CPC_UNI_PRCP/GAUGE_GLB/V1.0/. This dataset integrates all existing CPC information sources and employs the optimum-interpolation objective analysis technique to form a set of cohesive precipitation

products with consistent quantity and enhanced quality. The data was collected from > 30,000 monitoring stations belonging


to the WMO Global Telecommunication System, Cooperative Observer Network, and other national meteorological agencies (Xie et al., 2007). For the sake of brevity, the CPC-Global data is referred to as CPC in this article, and the corresponding SWAT model is denoted as the CPC model.

In the SWAT model, all meteorological data were categorized into sub-basins according to the "nearest distance" principle. As such, for the point-formatted gauge observations, a SWAT sub-basin will read the precipitation records from the weather station that is closest to its centroid. Similarly, for the grid-formatted estimates, i.e. CHIRPS and CPC, a SWAT sub-basin will read the precipitation observations from the grid that is closest to its centroid. Using this method, some of the grid records are potentially missed, especially for the high-resolution CHIRPS products. In order to incorporate the advantages of CHIRPS' spatial resolution and the SWAT model's effectiveness when using the other two products, we selected 400 sub-basins, so that the number of effective CHIRPS is ~ 20 times greater than that of the Gauge and CPC. Note that all of the precipitation statistics in this study are based on the sub-basin scale, which ensures that the precipitation data was correctly categorized in the SWAT model.

## 2.3. Statistical analysis method

The following indices were selected to statistically compare the three precipitation products:

(1) *CC* value. The correlation coefficient (*CC*) is a numerical measure (ranging from -1 to 1) of the linear statistical relationship between two variables, e.g., simulations and observations, with respect to strength and direction. The closer the absolute value of *CC* to 1, the higher the correlation between simulation and observation. *CC* is mathematically expressed as follows:

$$CC = \frac{\sum_{i=1}^{n}(Q_i - \bar{Q})(S_i - \bar{S})}{\sqrt{\sum_{i=1}^{n}(Q_i - \bar{Q})^2}\sqrt{\sum_{i=1}^{n}(S_i - \bar{S})^2}}, \tag{1}$$

Where *n* is the number of the time series; $Q_i$ and $S_i$ are measured values and estimated values (or simulated values), respectively; and $\bar{Q}$ and $\bar{S}$ are the mean values of the measured and estimated values (or simulated values), respectively.



(2) *STD* value: Standard deviation (*STD*) represents the discretization degree of the datasets. The *STD* of Gauge observations is used to normalize the OPPs' *STD*, and denoted as $STD_n$ to compare the dispersion of OPPs relative to Gauge. $STD_n$ values range from 0 to 1, and the optimal value is 0. The *STD* value is mathematically expressed as follows:

$$STD = \sqrt{\frac{1}{n}\sum_{i=1}^{n}\left(S_i - \overline{S}\right)^2},$$  (2)

$$STD_n = STD_{OPP} / STD_{Gauge},$$  (3)

Where $STD_{OPP}$ and $STD_{Gauge}$ are the *STDs* of OPPs and Gauge, respectively.

(3) *RMSD* value: Root mean square deviation (*RMSD*) is used to demonstrate the error between the OPPs and Gauge datasets. *RMSD* has a range from 0 to $+\infty$, and an optimal value of 0. The *RMSD* value is expressed as follows:

$$RMSD = \sqrt{\frac{1}{n}\sum_{i=1}^{n}\left[(Si - \overline{S}) - (Qi - \overline{Q})\right]^2},$$  (4)

(4) *POD* value: Probability of detection (*POD*) is a ratio that reflects the number of times OPPs correctly detected the frequency of rainfall relative to the total number of rainfall events. *POD* has a range from 0 to 1, and an optimal value of 1. It is mathematically expressed as follows:

$$POD = \frac{t_H}{t_H + t_M},$$  (5)

Where $t$ is the number of qualified data pairs; $H$ denotes when the Gauge and OPPs both detect a rainfall event; and $M$ represents when Gauge detects a rainfall event, but the OPPs do not.

(5) *FAR* value: False-alarm rate (*FAR*) represents the frequency that precipitation is detected using OPPs, but not detected using Gauge. *FAR* has a range from 0 to 1, and an optimal value of 0. The *FAR* value is mathematically expressed as follows:

$$FAR = \frac{t_F}{t_H + t_F},$$  (6)

Where $F$ denotes that the OPPs detected a rainfall event, while the Gauge does not.



## 2.4. Hydrological model

### 2.4.1 SWAT description

The process-based SWAT model is an all-inclusive, temporally uninterrupted, and semi-distributed simulation that was developed by the Agricultural Research Service of the United States Department of Agriculture (Arnold et al., 1998; Arnold

& Fohrer, 2005). The model's smallest simulation unit is the Hydrological Response Unit (HRU). Fields with a specific LULC and soil, which may be scattered throughout a sub-basin, are lumped together in one HRU. The model assumes that there is no interaction between HRUs in any one sub-basin. According to the water balance cycle, the HRU hydrologic process is first calculated and then summed to obtain the total hydrologic process of the sub-basin (Arnold et al., 1998; Arnold et al., 2012). Water balance, including precipitation, surface runoff, evapotranspiration, infiltration, lateral and base flow, and percolation

to shallow and deep aquifers, is mathematically expressed as follows:

$$SW_t = SW_0 + \sum_{i=1}^{t}(P - Q_{surf} - E_a - W_{seep} - Q_{lat} - Q_{gw}) \tag{7}$$

Where, $SW_t$ is the soil's water content at the end of period $t$ (mm); $SW_0$ is the soil water content at the beginning of period $t$ (mm); and $t$ is the calculation period length. $P$ = precipitation; $Q_{surf}$ = surface runoff; $E_a$ = evapotranspiration; $W_{seep}$ = the amount of percolation and by pass flow exiting the soil profile bottom; $Q_{lat}$ = lateral flow, and $Q_{gw}$ = base flow, including return flow from the shallow aquifer (GW_Q) and flow out from the deep aquifer (GW_Q_D)—all on day $i$ in (mm).

Surface runoff, lateral flow, and base flow add up to Water Yield (WYLD). SWAT uses the Soil Conservation Services-Curve Number method (SCS-CN) to simulate surface runoff. Surface runoff is mathematically expressed as follows:

$$\frac{F}{S} = \frac{Q_{surf}}{P - I_a} \tag{8}$$

Where $I_a$ is the initial loss (mm)—i.e., precipitation loss before surface runoff; $F$ is the final loss— i.e., precipitation loss after surface runoff is generated; $S$ is the maximum possible retention in the basin at that time (mm), and is the upper limit of $F$.

The WYLD from the sub-basin forms in the connected channel network, then, through routing water processes, enters the downstream reach segment and repeats the water balance process, eventually converging at the drainage outlet. This is the





watershed hydrological process simulated in SWAT that is shown in Fig. 2.

### 2.4.2 SWAT calibration and validation

In this study, the period of calibration and validation are set for 1999-2008 and 2009-2018, respectively. The two years

before both the calibration and validation periods are delineated as the warmup phase for the purpose of initializing model

state variables, e.g., soil moisture and groundwater concentration. The model is automatically calibrated using the Sequential

Uncertainty Fitting algorithm version 2 (SUFI-2) in the SWAT- Calibration and Uncertainty Program (SWAT-CUP). This

algorithm has been successfully applied in many related studies (Abbaspour et al., 2017; Shivhare et al., 2018; Tuo et al., 2018).

The model parameters and the initial calibration range are shown in Table 1. The parameters were selected based on literature

published by Arnold et al. (2012) and Tuo et al. (2016) and the official manual. In order to consider the impact of elevation on

precipitation and the fact that precipitation in the OPPs is horizontal, we introduced the Precipitation Lapse Rate (PLAPS)

parameter and divided each sub-basin into 10 Elevation Bands. Tuo et al (2016) demonstrated that this method is able to correct

the rainfall error caused by ignoring elevation and effectively improve the model's simulation performance. Furthermore, the

models were calibrated following three iterations of 1000 times each. Following each iteration, the SWAT-CUP generated a

fresh set of parameter ranges. This new set was used for the next iteration after considering the upper and lower bounds of the

physical meaning. The above method was repeated for the three different precipitation data sets. The Nash efficiency

coefficient (*NSE*) was used as the objective function to optimize the model calibration, and is mathematically expressed as

follows:

$$NSE = \frac{\sum_{i=1}^{n}(S_i-Q_i)^2}{\sum_{i=1}^{n}(Q_i-\overline{Q})^2},$$

(9)

The model performance was classified using the *NSE* values defined by Moriasi et al. (2007): unsatisfactory performance

($NSE \leq 0.50$), satisfactory performance ($0.50 < NSE \leq 0.65$), good performance ($0.65 < NSE \leq 0.75$) and very good performance

($0.75 < NSE \leq 1.00$). The three models' parameter range after 3000 iterations is shown in Table 2. *CC*, *NSE*, and Percentage

bias (*PBIAS)* were used to evaluate the model simulation results.



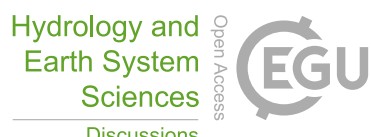

PBIAS describes the OPPs' systematic bias. *PBIAS* ranges from 0 to +∞, and the optimal value is 0. The calculation equation is expressed as follows:

$$PBIAS = \frac{\sum_{i=1}^{n}(S_i - Q_i)}{\sum_{i=1}^{n} Q_i} \times 100\%,$$  (10)

## 3. Results

### 3.1. Evaluation of the OPPs on a temporal scale

#### 3.1.1. Monthly scale

A comparison of the monthly precipitation time series (Gauge, CHIRPS, and CPC) across the watershed is shown in Fig. 3. Note that the time series in Fig. 3 represents the average value of the whole watershed and was calculated as follows: the original point or grid formatted rainfall records were first categorized into every SWAT model sub-basin, according to the nearest distance principle, and then spatially synthesized into one time series by sub-basin area using the weighted average method. Fig. 3 shows that the extreme rainfall values captured by Gauge are higher than those of CHIRPS and CPC. Moreover, the *CC* values between CHIRPS and Gauge records, as well as CPC and Gauge records, are 0.97 and 0.98 (P<0.01, i.e., extremely significant positive correlation), respectively. The high *CC* values demonstrate the highly correlated linear relationship between the two OPPs and Gauge records on a monthly scale, indicating that both CHIRPS and CPC products are equally as effective at describing the monthly precipitation variation within the JRW as the Gauge records.

The box diagrams of the three precipitation records are shown in Fig. 4. Note that July is the largest contributor to the yearly precipitation, as well as the annual flood peak calculation. According to the July results, when compared with Gauge records, the CHIRPS product has a large median, small maximum, and large minimum, while the three CPC values are all smaller than the Gauge values. These characteristics will potentially lead to different hydrological modeling in flood peak simulation. The *STD* values for Gauge-CHIRPS and Gauge-CPC are 1.06 and 0.94, respectively. The *RMSD* values for Gauge-CHIRPS and Gauge-CPC are 15.80 and 12.95, respectively. These statistics indicate that both CHIRPS and CPC estimates are

able to provide equally effective precipitation values compared with that of the Gauge records. Nevertheless, *PBIAS* values of Gauge-CHIRPS and Gauge-CPC were 9.58 and -6.70, respectively, indicating the overestimation of CHIRPS products and underestimation of CPC products compared with that of the Gauge records. Specifically, overestimation of the CHIRPS products mainly occurs between April and September, which is the JRW rainy season; while during the dry season, i.e., October - March, the CHIRPS estimates are closely consistent with the Gauge records. In contrast, the CPC estimates parallel the Gauge records for the rainy season, yet rainfall is underestimated during dry season.

### 3.1.2. Daily scale

Intensity and frequency are the most critical parameters for characterizing rainfall features on a daily scale (Azarnivand et al., 2019; Wen et al., 2019). The scatter plots in Fig. 5 depict a precipitation intensity comparison between the OPPs and Gauge records, at a daily scale, at Beibei hydrological station (NO.411). Based on Fig. 5, the angle between the CHIRPS' 95 % line estimates and the horizontal axis is > 45 degrees (the 1:1 line), indicating that CHIRPS overestimates precipitation relative to the Gauge records. CPC estimates demonstrate the exact opposite. More specifically, the scatter distribution indicates that compared to the Gauge records, the CHIRPS' estimates tend to overestimate precipitation for light rains, and underestimate it for heavy rains. Meanwhile, the CPC products underestimate both light and heavy rains. Statistically, the *CC*, *STD,* and *RMSD* values between CHIRPS and the Gauge records are 0.53, 1.14, and 5.16, respectively, and 0.64, 0.87, and 3.95, respectively, between the CPC and Gauge products. At the daily scale, the OPPs and Gauge products showed an evident decrease in consistency when compared with the monthly scale. The above indicators demonstrate the deceased consistency of the OPPs' estimates and Gauge observations; while the CPC shows superior performance relative to the CHIRPS.

The three precipitation products' cumulative daily precipitation intensity frequencies are shown in Fig. 6. Note that on the right side of the figure, the 50 mm/day demarcation divides the horizontal axis into two sections of rainfall intensity, in order to depict the three products' frequency trends more clearly. Overall, the three products display a high probability of occurrence for precipitation intensity of 0.1~25 mm/day—87 %, 94 %, and 98 % for CHIRPS, Gauge, and CPC, respectively. However, the probability for precipitation intensity > 100 mm/day is 99.70 %, 99.73 % and 99.99 %, respectively, indicating the potential upper limit extreme rainfall event value within this area. The CPC product fails to detect extreme rainstorm events.

Table 3 is the recognition capability evaluation of the two OPPs for rainfall intensity between 0.1 and 50 mm and > 50 mm events. The CPC and CHIRPS *POD* values for rainfall intensity between 0.1 and 50 mm are 83.53 % and 27.29 %, respectively, demonstrating that the CPC product has a strong ability to capture the onset of rainfall. Nevertheless, the CPC

*POD* value for rainfall intensities≥ 50mm decreases to 9.42 %, indicating its poor ability to capture rainstorms, while that of the CHIRPS product is relatively higher, with a *POD* value of 18.12 %. Moreover, the *FAR* fractions for both OPPs are between 44 % and 66 %, demonstrating its lower ability to detect rainstorm values.

### 3.2. Evaluation of OPPs on a spatial scale

Spatial varations of the three products' long-term mean annual precipitation, for all of the partitioned sub-basins, are

shown in Fig. 7. The three products' precipitation values exhibit an obvious upward trend from the JWR's upstream to downstream region. The precipitation's shifting pattern in space highly correlates with the topography variation (shown in Fig. 1), indicating that meteorological and hydrologic variables throughout the region are potentially influenced by and respond to catchment landscape modification. It should be noted that in Fig. 7(a), all the sub-basins are divided into several regions, each of which has the same rainfall value, while abrupt rainfall value changes occurred between adjacent regions. In Fig. 7(c), the

transition between adjacent sub-basins is smoother than that observed with Gauge. In Fig. 7(b), the CHIRPS product shows the smoothest precipitation transition between the adjacent sub-basins, illustrating the advantages of the high resolution CHIRPS product. In this study, precipitation records from 20 rain gauge stations, 411 CHIRPS grids, and 76 CPC grids were categorized into the SWAT sub-basins (as mentioned in section 2.3), leading to differences in the continuity or smoothness of rainfall spatial distribution among the three products. Compared with the Gauge observations, the overall precipitation values

estimated by the CHIRPS are relative higher, while that of the CPC is relatively lower.

The correlation coefficients' spatial variation between the Gauge and OPPs at monthly and daily scales are illustrated in Fig. 8. Overall, the monthly scale *CC* values (with a rang of 0.7~1) are comparably larger than that of the daily scale values (with a rang of 0.5~0.7). Spatially, the higher *CC* values between the Gauge and CPC at the monthly scale are mainly distributed in areas with comparably low or high rainfall amounts, such as Wudu and Wangyuan. Yet, the *CC* value was less

relevant in areas with moderate rainfall (e.g., Suining) relative to that of the Gauge and CHIRPS. However, at the daily scale,



the correlation of Gauge and CPC is higher than that of Gauge and CHIRPS, except for a few individual sub-basins located in the east-south area.

## 3.3. Hydrological performance of different precipitation products in the SWAT model

### 3.3.1 Spatio-temporal performance at a monthly scale

OPPs ignores terrain differences when forcing the model, which may increase potentially systematic errors in hydrologic modeling (Tuo et al., 2016). Thus, in this work, elevation bands (see Sect. 2.4.1) were used to normalize precipitation at different elevations. The monthly observed runoff and simulated runoff subjected to this procedure, and used for the SWAT model during the calibration and validation periods, are presented in Fig. 9. The results show that the three precipitation inputs successfully stimulated the model to reproduce the discharge records at the Beibei station; the rising and falling simulated

flood event processes are in good agreement with that of the observed ones. Based on the model performance classification scheme designed by     Moriasi et al. (2007), all three models, each using a different precipitation product, achieved "very good" performance for both the calibration and verification periods, although the Gauge model attained the highest *CC* (0.93 for calibration and 0.87 for validation) and *NSE* (0.92 and 0.87). Compared with the model using Gauge input, the models using the two OPPs tended to underestimate the peak flows that occur mainly during flood seasons (June to August), which is the

main reason behind the lower *NSE* values. The Gauge model demonstrates the best performance, which may reflect its strong ability to ascertain the peak rainfall during the flood seasons (Fig. 4). Note that in Fig. 4, the CHIRPS medians are larger than that of the Gauge, while the maxima are smaller, and the minima are larger during the flood seasons. These features facilitated the best performance for describing the base flow and medium floods, like those in years 2003 and 2014. As a result, the CHIRPS model achieved the best simulation base flow ; although it overestimated the precipitation with light rain intensity,

and obviously overestimated the streamflow with discharge< 6000m³/s, which also led to its final performance deviation. CPC showed significant overestimation in 2017 and 2018 during the verification period. Although it approximated CHIRPS' estimated results, it clearly deviated from its previous tendency to underestimate precipitation during these two years.

In terms of simulated WYLD spatial variation at the sub-basin scale (as shown in Fig. 10), the consistency of the Gauge and CHIRPS models is slightly better than that of the CPC model, potentially demonstrating the advantage of the CHIRPS'

high resolution in simulating precipitation. Furthermore, the WYLD distribution pattern is highly consistent with the

corresponding precipitation distribution (Fig. 7). The spatial correlation between WYLD and precipitation for rainfall for the

Gauge, CHIRPS, and CPC products reached 84.8 %, 84.3 %, and 90.84 %, respectively. Compared to the Gauge simulation,

the CHIRPS overestimated and the CPC underestimated the WYLD. The *PBIAS* values for Gauge-CHIRPS and Gauge-CPC

are 5.85 and -5.38, respectively.

**3.3.2. Spatial-temporal performance at a daily scale**

As shown in Fig. 11, the three precipitation inputs also successfully forced the model to replicate the discharge records

at the Beibei station at a daily scale, with performance evaluations of "good," "satisfactory," and "satisfactory" for Gauge,

CHIRPS, and CPC models, respectively. The performances in describing the peak flows are not very good for all of the three

products, among which, the Gauge model performs best. The peak flows are usually caused by extreme precipitation events,

like rainfall events with an intensity > 80 mm/day. As shown in Figs. 5 and 6, both the CHIRPS and CPC underestimate heavy

rainfall intensities compared with the Gauge observations. Conversely, the CHIRPS model performs best in simulating the

base flow, since CHIRPS tend to capture higher values of light rainfalls than that of the Gauge and CPC.

Note that at a daily scale, the three evaluation parameters are significantly smaller than at the monthly scale. The primary

reason is that the daily scale sample size is nearly 30 times larger than that of the monthly scale, so the cumulative systematic

deviation led to poorer evaluation parameters. Therefore, in order to better compare the simulation results of the two scales,

the streamflow discharge daily process was integrated into the monthly process, so that it had the same sample size as the

monthly model. As shown in Fig. 12, the three types of precipitation models maintained their inherent advantages and attained

equal or superior performance at the daily scale. The WYLD spatial variation for all sub-basins at the daily scale are shown in

Fig. 13. Similar to the results from the monthly scale, the WYLD spatial pattern is highly correlated with that of precipitation.

The *CC* values between the WYLD and precipitation for Gauge, CHIRPS, and CPC at the daily scale are 83.45 %, 84.41 %

and 91.70 %, respectively, which are even higher than those of the monthly scale. The *CC*, *STDn*, and *RMSD* values between

CHIRPS and Gauge are 0.92, 1.06, and 0.23, respectively, and 0.81, 0.94, 0.33 between CPC and Gauge, respectively.



## 4. Discussion

### 4.1. Comparison of precipitation products in terms of rainfall events with different magnitudes

365       In the aforementioned results, compared to Gauge product, the CHIRPS tends to overestimate the intensity and frequency

of light rain, but underestimate heavy rain, which is consistent with the results reported by Gao et al. (2018). However, CPC

tends to underestimate the intensity and overestimate the frequency of rainfall for light and heavy rain, although light rain is

more underestimated. These results are consistent with those reported by Ajaaj et al. (2019). The differences in the capture of

different magnitude rainfall intensities may potentially influence hydrologic process and forecasting. From Eq. (8), the basin's

WYLD is directly proportional to the amount of precipitation. In other words, heavy rainfall tends to produce large amounts

of streamflow. Duan J et al. (2019) conducted a slope experiment and found that there was a significant difference in the runoff

coefficients between extreme rainfall events and normal rainfall events; the former produced much more runoff and sediment

than the latter. Solano-Rivera et al. (2019) experimented in the San Lorencito headwater catchment and found that the rainfall-

runoff dynamics before extreme events were mainly related to early-stage conditions. After extreme flood events, early-stage

conditions had no effect on rainfall-runoff processes, and rainfall significantly affected the streamflow discharge. Moreover,

the evaluation index *NSE* performance is mainly determined by the peak streamflow. Thus, it is critical to identify the

magnitudes of different rainfall events at both a temporal and spatial scale. As a consequence, we derived the temporal and

spatial distributions of the rainfall events with different magnitudes. The spatial scale dimension was implemented by

identifying the serial numbers of all sub-basins (Fig. 14); and the temporal dimension was fulfilled by detecting rainfall events

of different magnitudes throughout the study period (Fig. 15).

      Overall, Fig.15 shows that the CPC tends to capture more light rainfall events with precipitation intensities between 0.1

and 50 mm/day (LR events), the CHIRPS identified more medium rainfall events with precipitation intensities between 50 and

100 mm/day (MR events), and both the Gauge and CHIRPS detected more heavy rainfall events with precipitation intensities

larger than 100 mm/day (HR events). Accordingly, the total annual precipitation amounts of the three products are ranked as

CHIRPS (956.4 mm) > Gauge (872.8 mm) > CPC (814.3 mm). Even with the advantage of detecting MR and HR events, the

CHIRPS' ability to simulate flood events is inferior to that of the Gauge. Potential reasons may consist of: 1) the HR events

detected by CHIRPS are more scattered at a temporal scale, which disperses the flood peak value; and 2) the high frequency

of the MR detected by CHIRPS resulted in parameter sets in the SWAT model that tended to derive a lower runoff coefficient,

in order to avoid a large systematic bias in terms of *PBIAS*.

The CPC estimates, 60% of which are detected as LR, tend to be incapable of driving the SWAT model to capture small

streamflow discharge, especially the ones equivalent to base flow. Consequently, the CPC model *CC* values are relatively low.

A potential reason for this phenomenon may be that the rainfall during LR events tends to be easily lost in the initial- and post-

loss processes, resulting in low proletarian flow and thus WYLD. Furthermore, the CHIRPS has a high probability of MR

event false-alarm, which is consistent with the results reported by Zambrano-Bigiarini et al. (2017). Thus, a significant number

of erroneous peaks exist in the CHIRPS, just like the temporal variation at a daily scale, which has a very low correlation with

the Gauge. Erroneous precipitation peaks tend to produce erroneous streamflow peaks. Although SWAT can repair the peak

position deviation to some extent, the *CC* is inevitably reduced.

### 4.2. Effect of OPPs difference on hydrological process simulation

In general, simulated and observed streamflow hydrographs, using OPPs and Gauge inputs, can successfully match at

both monthly and daily scales. However, consistency between simulated and observed streamflow does not guarantee identical

hydrologic processes. For example, the SWAT model calibrated parameters are not the same for all precipitation inputs,

meaning that the hydrologic mechanics during SWAT modeling are also different. As such, it is critical that researchers and

decision makers adequately understand the benefits and limitations of different precipitation products in modeling the

hydrologic processes.

According to the SWAT model's water balance equation (Eq. 9), WYLD equals the sum of $Q_{surf}$, $Q_{lat}$ and $Q_g$, where $Q_{gw}$

can be divided into flow out from a shallow aquifer (GW_Q) and flow out from a deep aquifer (GW_Q_D). If the soil water

content $W$ and percolation/bypass flow into the deep aquifer $w_{seep}$ remains unchanged over a long time period, then the equation

is modified to $P= Q_{surf} + E_a + Q_{lat} + Q_{gw}$. Thus, we calculated the water balance component portions, $Q_{surf}$, $Q_{lat}$, $Q_{gw}$, and $E_a$,

for all the JRW sub-basins. It is evident from Fig.16 and Table 4 that the total portions of water balance components differ

among the three precipitation products. However, they do share some similarities in that the evapotranspiration (ET) portions

of all three products are above 50 %, resulting in a watershed runoff production coefficient of ~0.45. Furthermore, the main Gauge model components are SURQ and LATQ, which account for 25.92 % and 16.72 %, respectively; the main CHIRPS component is SURQ, which accounts for 34.80 %, and the main CPC component is LATQ, which accounts for 33.62 %. Spatially, the surface flow portion increases from upstream to downstream.

415       The above water balance component regularities are primarily the result of two causes. First, the differences in the above hydrological component proportions are mainly controlled by the model parameters. For example, ESCO is a soil evaporation compensation factor that directly affects maximum evaporation from soil; the smaller the value, the larger the maximum evaporation. The SWAT model indirectly increases WYLD by using higher ESCO and thus decreases the ET value. In this study, the ESCO values for Gauge, CHIRPS, and CPC range from 0.879 - 1, 0.775 – 1, and 0.914 - 1, respectively. Furthermore,

the total ET values during the study period were 8153.94, 8161.22, and 7806.84 mm, respectively. Apparently, the CPC model reduced its corresponding ET by using a higher ESCO parameter, so that the lack of precipitation inputs would be offset by less evaporation. This result is consistent with that reported by Bai & Liu (2018), who conducted a study at the source regions of the Yellow River and Yangtze River basins in the Tibetan Plateau. They further concluded that the impact of different precipitation inputs on runoff simulation is largely offset by parameter calibration, resulting in significant differences in

evaporation and storage estimates.

      Second, rainfall characteristics also have a significant impact on hydrological processes in the watershed. A large number of studies show that rainfall intensity is a key player in the watershed's hydrological process (Zhou et al., 2017; Du et al., 2019; Zhang et al., 2019). Studies conducted by Redding & Devito (2010) showed that the occurrence of lateral flow is mainly determined by rainfall intensity. When the rainfall intensity is greater than the surface soil hydraulic conductivity, the rainfall

mainly forms surface runoff. When rainfall intensity is between the soil surface and bedrock hydraulic conductivity, the rainfall mainly forms lateral flow. When rainfall intensity is less than the bedrock's hydraulic conductivity, the rainfall will infiltrate into the groundwater. In this study, the precipitation recorded by CHIRPS was mainly distributed between 25 and 100 mm/day, while that of CPC was mainly distributed between 0.1 and 25 mm/day. This may be the reason why CHIRPS overestimated the proportion of surface runoff and CPC overestimated the proportion of lateral flow, compared with that of the Gauge model.

Moreover, precipitation in the watershed's upstream area tended to infiltrate into the land surface due to the lower precipitation

detection (see Fig. 7); yet when the river flow converged in the watershed's downstream area, the surface flow increased due to the larger detected precipitation values.

## 5. Conclusions

The sparsity and unevenness of ground-based precipitation observations pose great challenges to the establishment of
hydrological models. In this study, the Gauge, CHIRPS, and CPC were evaluated by statistically comparing the different precipitation products as well as their performance at driving the hydrological model. Specifically, the potential behaviors of different precipitation datasets in describing precipitation magnitudes and hydrologic processes in terms of water balance components are further discussed. The main conclusions are summarized as follows:

1. The three precipitation datasets exhibited similar temporal records at a monthly scale, and the Gauge measures were
more capable of capturing maxima than those of the OPPs. During rainy seasons, the CHIRPS median, maxima, and minima were larger, smaller, and larger, respectively, than that of the Gauge records, while all three CPC statistical values were smaller than that of the Gauge. At a daily scale, the CHIRPS tends to overestimate light rains and underestimate heavy rains, while all the CPC rainfall intensities were underestimated. Spatially, precipitation in all sub-basins increases from the upstream to the downstream region, and the CHIRPS derives the most smoothly distributed precipitation pattern.

2. All three precipitation inputs successfully forced the model to replicate the discharge records at the Beibei station at a monthly and daily scale, although they performed slightly better at the daily scale. The differences in the statistics at the monthly and daily scale correspondingly affected the streamflow photographs, e.g. flood peak, base flow, and the rising and falling processes. The three models' spatial WYLD distributions are highly correlated to that of the precipitation records. While there were equivalent performances in simulating streamflow hydrographs, it should be noted that the calibrated parameters in
all three models (Gauge, CHIRPS, and CPC models at monthly and daily scales, see Table 2) were quite different. In other words, evaluating only the streamflow simulation accuracy of the precipitation products will conceal the differences between these precipitation products, which is primarily because that hydrological models are able to offset the influences of precipitation inputs on streamflow simulations using parameter calibration and validation.



3. The calibrated parameters are adjusted to alter the hydrologic mechanics in terms of water balance components. Thus, they effectively fill the potential gaps in the WYLD that may be introduced by the varying precipitation amounts and intensities detected by different precipitation products. In particular, according to parameter adjustment, the three products' precipitation detection features resulted in significantly different water balance component portions, i.e., the overestimation of MR by CHIRPS resulted in a larger portion of surface flow, while the underestimation of all rainfall by CPC reduced a larger portion of lateral flow. Lastly, the spatial precipitation pattern also significant impacted the spatial distribution of the water balance components from upstream to downstream.

Although the OPPs have advantages and limitations with respect to the accuracy of precipitation estimates at different spatial and temporal scales, as well as in hydrological modeling and describing hydrologic mechanics, they demonstrate good potential in our case study within the JRW. As such, the OPPs should merge the advantages of satellite, ground observations, as well as the reanalyzed data. Furthermore, fully consideration on performing the hydrological evaluation from both spatial and temporal scales is also key for the future development of OPPs.

**Data availability**

Sources of the geospatial and climate forcing data used to configure the SWAT model have been described in the "2.2. Data sources". The simulations of streamflow data are shown in the figures of the paper and are available through contacting the authors.

**Author contributions**

HZ conceptualized the work. HZ and JP designed the core structure and collected the data required. JP conducted the data analyses and model simulation and drafted the manuscript. HZ revised the manuscript. QX collected data and contributed to key analyses and discussions. YJW, YQW and OZ contributed to the discussion of the results, and JH assisted carrying out model configuration and data analyses.



**Competing interests**

The authors declare that they have no conflict of interest.

**Acknowledgements**

This study was supported by the Fundamental Research Funds for the Central Universities (NO.2016ZCQ06 and
NO.2015ZCQ-SB-01), the National Natural Science Foundation of China (51309006, 41790434), and the National Major
Hydraulic Engineering Construction Funds "Research Program on Key Sediment Problems of the Three Gorges Project"
(12610100000018J129-01). We gratefully acknowledge the Beijing Municipal Education Commission for their financial
support through Innovative Transdisciplinary Program "Ecological Restoration Engineering". We also sincerely thank
anonymous reviewers and editor for their constructive comments and suggestions.

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



**Table 1: Hydrological parameters considered for sensitivity analysis ("a_", "v_" and r_" means an absolute increase, a replacement, and a relative change to the initial parameter values, respectively).**

| Parameters | Description | Range | Default |
|---|---|---|---|
| v__PLAPS.sub | Precipitation lapse rate[mm] | -1000/1000 | 0 |
| a__SOL_K().sol | Saturated hydraulic conductivity [mm/h] | −10/10 | Soil layer specific |
| r__SOL_BD().sol | Moist bulk density [g/cm$^3$] | −0.5/0.5 | Soil layer specific |
| a__CN2.mgt | SCS streamflow curve number | −20/20 | HRU specific |
| v__ESCO.hru | Soil evaporation compensation factor | 0/1 | 0.95 |
| a__HRU_SLP.hru | Average slope steepness [m/m] | −0.2/0.4 | HRU specific |
| a__SLSUBBSN.hru | Average slope length [m] | −9/130 | HRU specific |
| v__CH_K2.rte | Effective hydraulic conductivity [mm/h] | 0/400 | 0 |
| v__CH_N2.rte | Manning's n value for main channel | 0/0.3 | 0.014 |
| a__GWQMN.gw | Threshold depth of water in the shallow aquifer required for return flow to occur [mm] | −500/500 | 1000 |
| a__REVAPMN.gw | Threshold depth of water in the shallow aquifer for "revap" to occur [mm] | −500/500 | 750 |
| v__GW_REVAP.gw | Groundwater "revap" coefficient | 0.02/0.2 | 0.02 |
| v__GW_DELAY.gw | Groundwater delay [days] | 0/300 | 31 |
| v__ALPHA_BNK.rte | Baseflow alpha factor for bank storage | 0/1 | 0 |




**Table 2: Optimal parameters calibrated for all three models.**

| Parameters | Initial range | Gauge | | CHIRPS | | CPC | |
|---|---|---|---|---|---|---|---|
| | | Monthly | Daily | Monthly | Daily | Monthly | Daily |
| v__PLAPS.sub | −1000/1000 | 0.012/0.067 | 0.061/0.183 | 0.079/0.135 | 0.068/0.205 | 0.017/0.078 | -0.014/0.095 |
| a__SOL_K().sol | −10/10 | 1.988/10 | -0.706/10 | -0.471/7.681 | -0.396/10 | 5.264/10 | -2.106/10 |
| r__SOL_BD().sol | −0.5/0.5 | 0.036/0.5 | -0.111/0.5 | -0.130/0.5 | -0.126/ 0.5 | 0.262/0.5 | -0.04/0.5 |
| a__CN2.mgt | −20/20 | -16.141/17.309 | -1.371/20 | 12.825/20 | -1.491/20 | -4.092/20 | -1.992/20 |
| v__ESCO.hru | 0/1 | 0.879/1 | 0.405/1 | 0.775/1 | 0.355/1 | 0.914/1 | 0.462/1 |
| a__HRU_SLP.hru | −0.2/0.4 | 0.261/0.4 | 0.013/0.4 | 0.157/0.280 | -0.2/0.116 | 0.181/0.4 | 0.049/0.4 |
| a__SLSUBBSN.hru | −9/130 | -9/40.518 | -9/75.760 | 68.303/108.959 | 23.139/94.386 | -9/19.244 | -9/74.023 |
| v__CH_K2.rte | 0/400 | 0/101.266 | 0/252.317 | 0/113.457 | 56.486/285.514 | 16.056/326.448 | 0/220.314 |
| v__CH_N2.rte | 0/0.3 | 0.019/0.188 | 0/0.173 | 0.091/0.183 | 0/0.187 | 0.138/0.233 | 0.091/0.272 |
| a__GWQMN.gw | −500/500 | -500/-241.312 | -76.285/500 | -104.708/178.914 | -500/21.287 | -500/-235.592 | -500/118.785 |
| a__REVAPMN.gw | −500/500 | -500/-98.63 | -500/125.78 | -429.291/69.66 | -232.285/303.28 | -189.739/295.43 | -432.284/189.28 |
| v__GW_REVAP.gw | 0.02/0.2 | 0.038/0.141 | 0.02/0.123 | 0.127/0.188 | 0.02/0.124 | 0.037/0.141 | 0.077/0.192 |
| v__GW_DELAY.gw | 0/300 | 37.681/215.552 | 118.714/300 | 96.336/188.942 | 81.664/245.036 | 0/74.581 | 0/182.936 |
| v__ALPHA_BNK.rte | 0/1 | 0.492/0.863 | 0.444/1 | 0.201/0.696 | 0.467/1 | 0.564/1 | 0.307/0.92 |





**Table 3: *POD* and *FAR* values for different rainfall intensities.**

|  | > 0.1mm | | ≥ 50mm | |
| --- | --- | --- | --- | --- |
|  | *POD* | *FAR* | *POD* | *FAR* |
| CHIRPS | 27.29% | 54.12% | 18.12% | 65.56% |
| CPC | 83.53% | 46.76% | 9.42% | 44.71% |





**Table 4: Summarization of water balance components of the three models for the whole JRW.**

| Datasets | Statistics | SURQ | LATQ | GW_Q | GW_Q_D | ET | Summation |
|---|---|---|---|---|---|---|---|
| Gauge | Average | 3832.20 | 2471.80 | 274.05 | 54.48 | 8153.94 | 14786.47 |
| | Percentage | 25.92% | 16.72% | 1.85% | 0.37% | 55.14% | |
| CHIRPS | Average | 5188.42 | 629.06 | 809.19 | 120.64 | 8161.22 | 14908.52 |
| | Percentage | 34.80% | 4.22% | 5.43% | 0.81% | 54.74% | |
| CPC | Average | 910.37 | 4707.44 | 547.85 | 28.32 | 7806.84 | 14000.82 |
| | Percentage | 6.50% | 33.62% | 3.91% | 0.20% | 55.76% | |






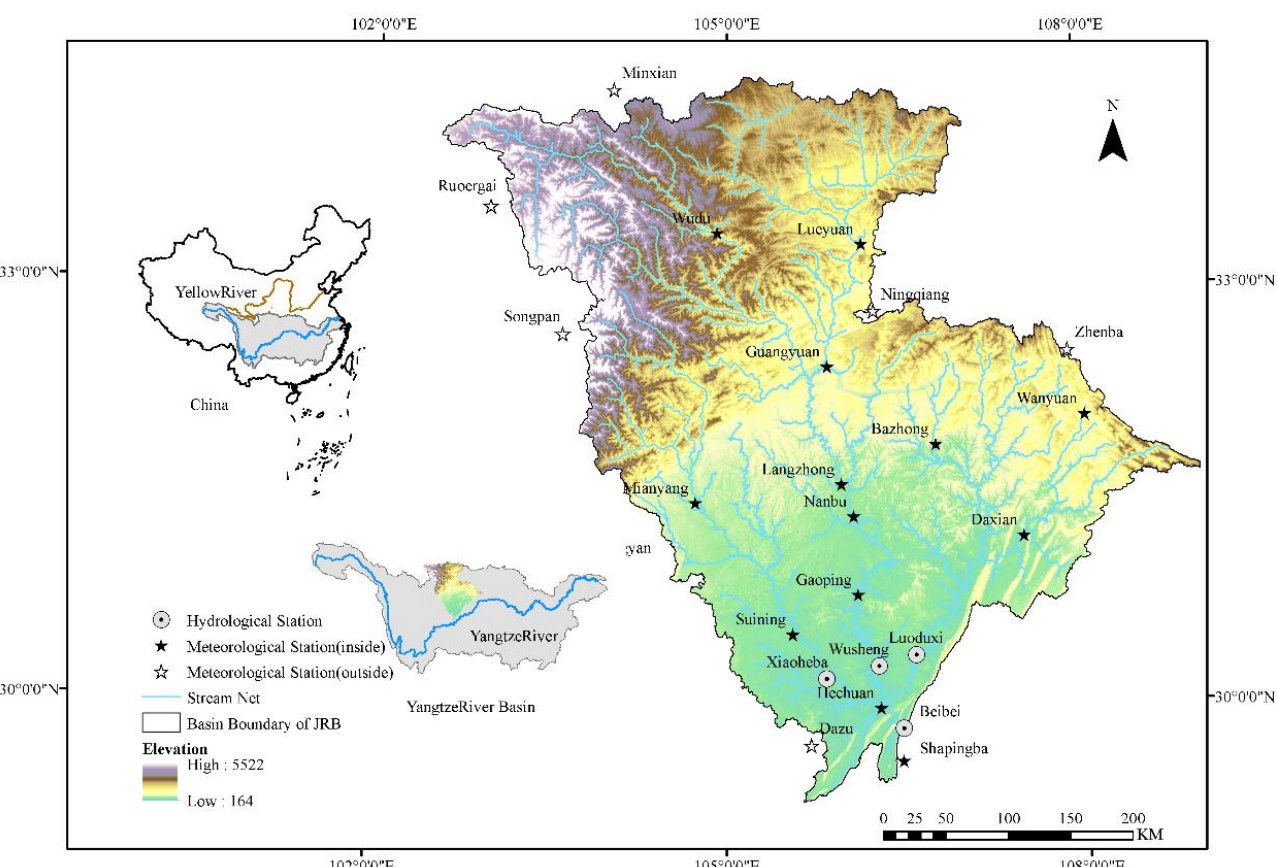

**Figure 1.** Sketch map of the Jialing River Basin with meteorological stations.



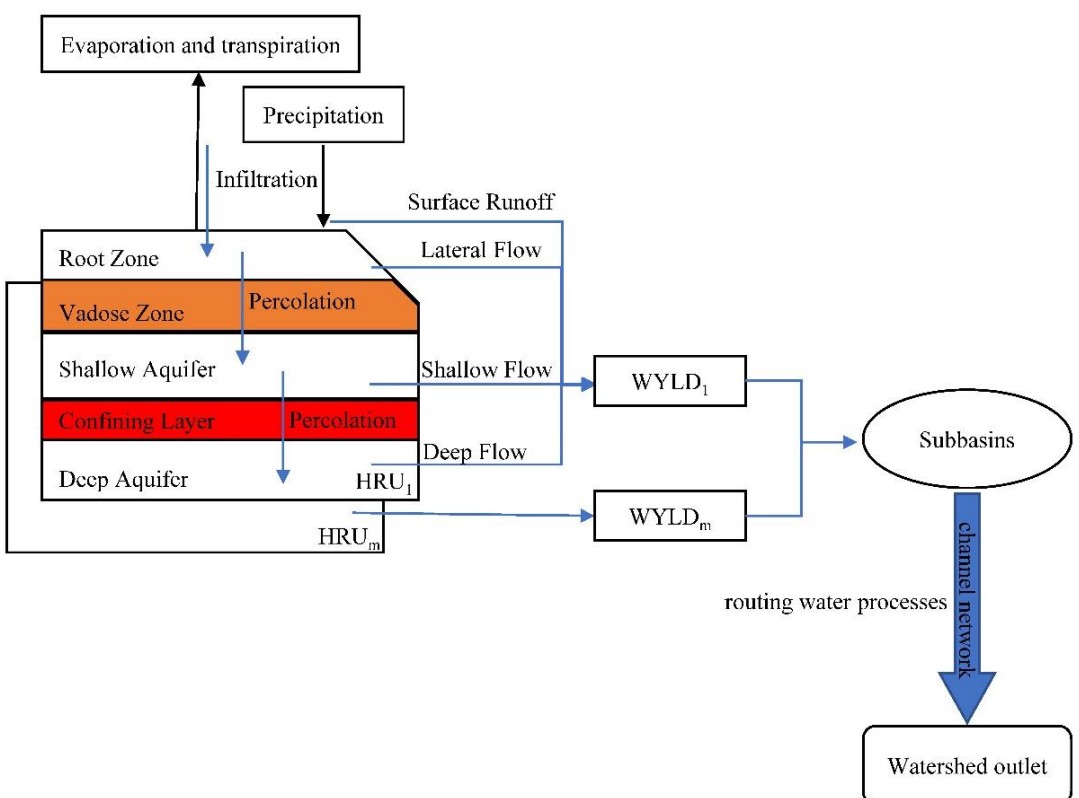

**Figure 2.** Schematic representation of the modified SWAT model structure.


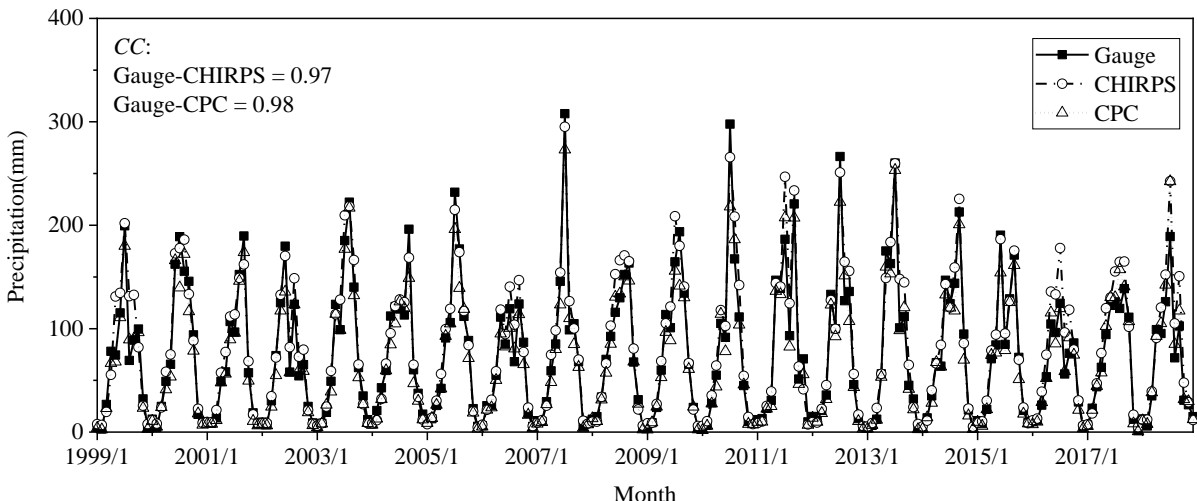

**Figure 3.** Time series of three different precipitation records at monthly scale in JRW (the *CC* values of 0.97 and 0.98 indicating extremely significant positive correlation (*P*<0.01, where *P* stands for the probability of being rejected when there was a significant difference)).




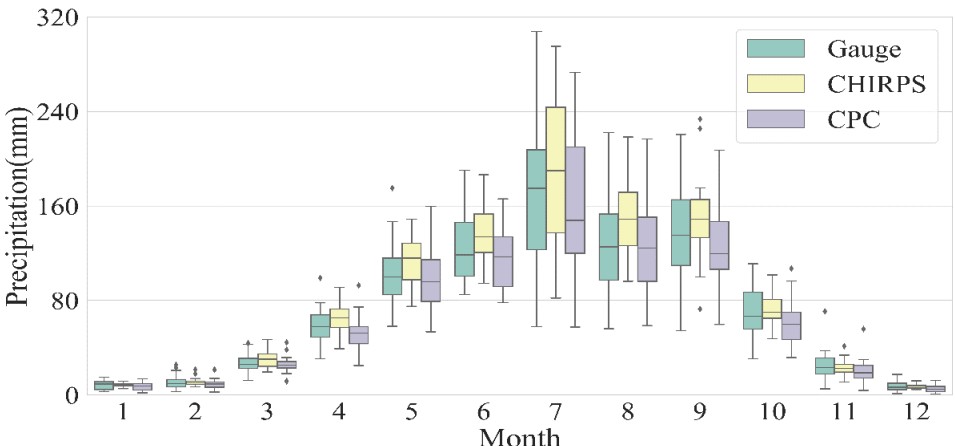

**Figure 4.** Box Diagrams of three different precipitation records at monthly scale in JRW.



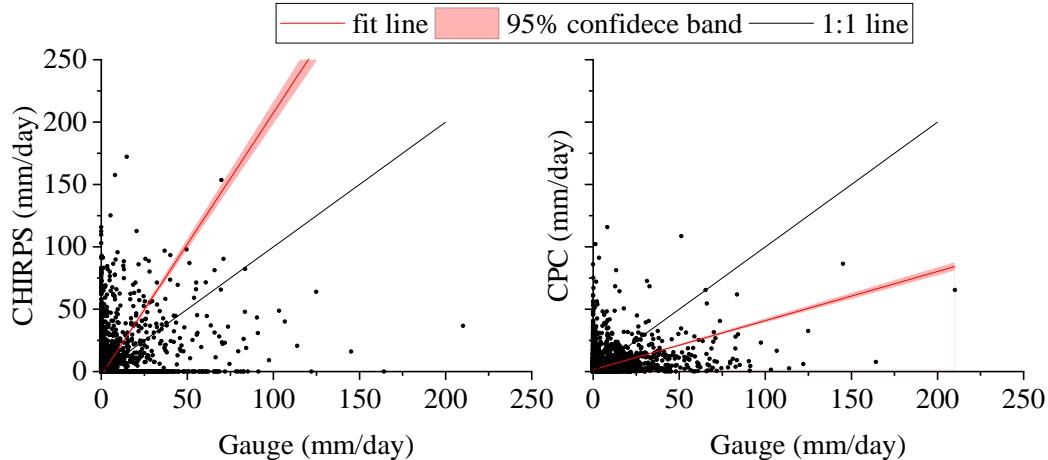

**Figure 5.** Scatter plot of the OPPs records comparing with Gauge records at daily scale: (a) comparison of CHIRPS and Gauge; (b) comparisons of CPC and Gauge.



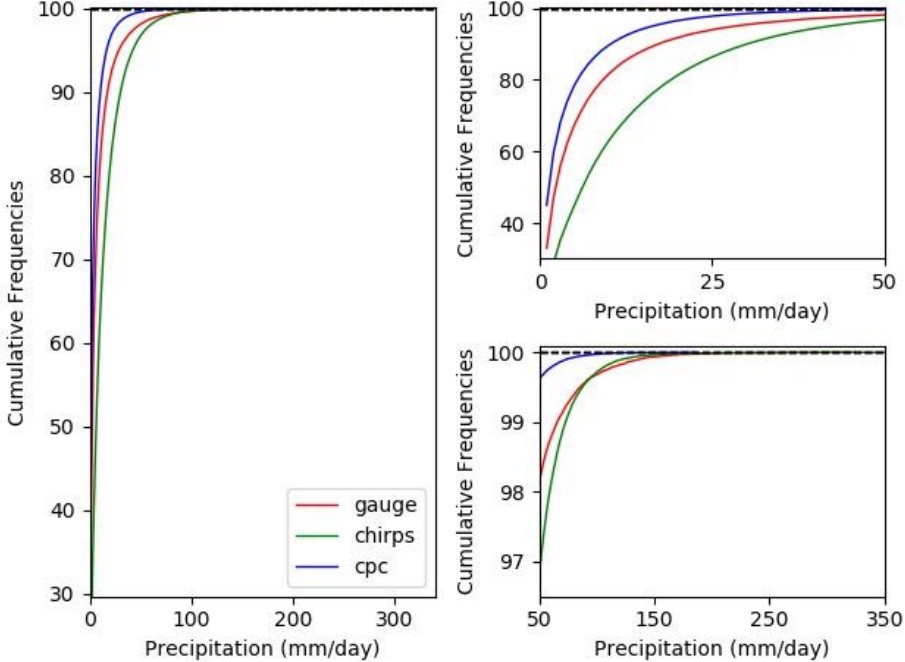


**Figure 6.** Cumulative Frequencies of daily precipitation intensity for the three precipitation products (Gauge, CHIRPS, CPC) in JRW: (a) distribution of all precipitation values; (b) distribution of precipitation values that are<100 mm; (c) distribution of precipitation values that are ≥100 mm.






**Figure 7.** Spatial variation of annual precipitation at sub-basin scale for (a) Gauge (b) CHIRPS and (c) CPC.


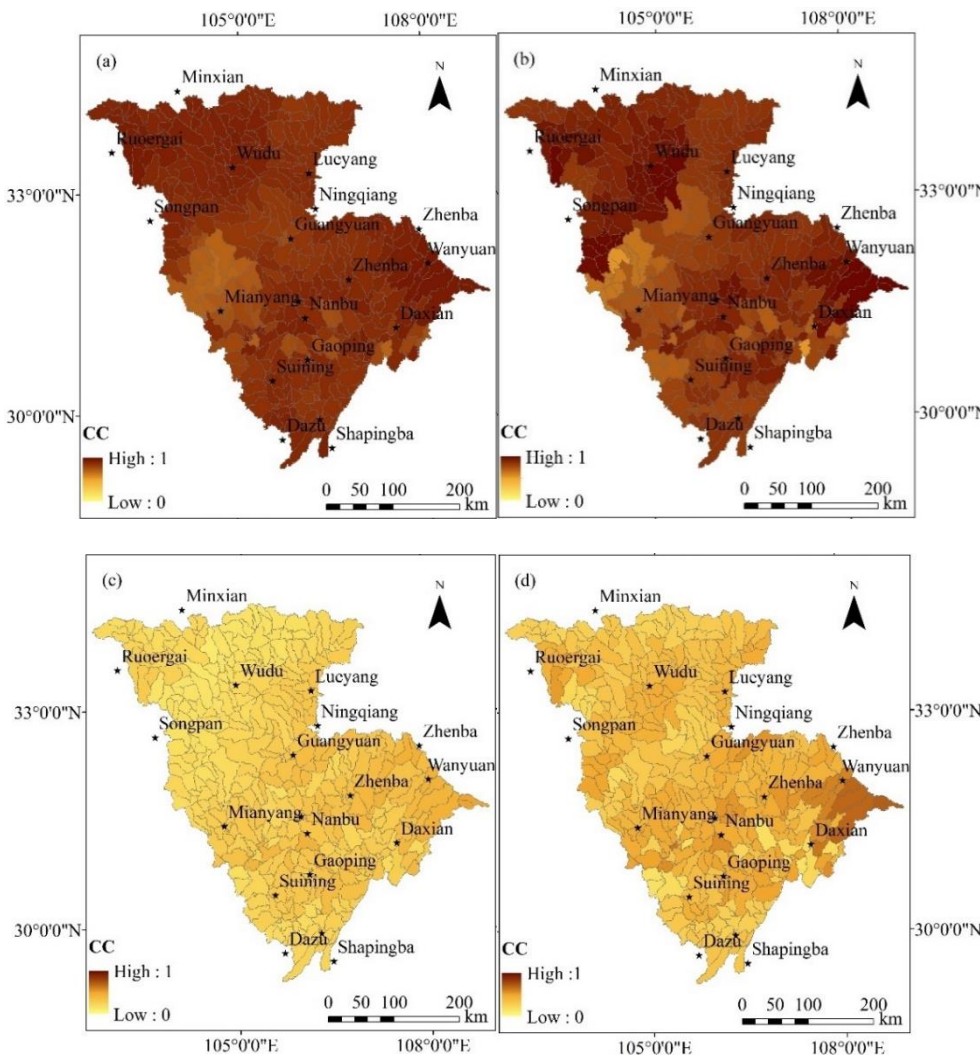


**Figure 8.** Spatial variation of *CC* values of the precipitation between (a) Gauge and CHIRPS, (b) Gauge and CPC at monthly scale and (c) Gauge and CHIRPS, (d) Gauge and CPC at daily scale.



**Figure 9.** Observed and simulated discharges at the outlet of JRW at monthly scale using precipitation inputs of Gauge, CHIRPS and CPC, respectively.





**Figure 10.** Spatial variation of water yield at monthly scale for all sub-basins calculated with precipitation inputs of (a) Gauge (b) CHIRPS and (c)CPC.





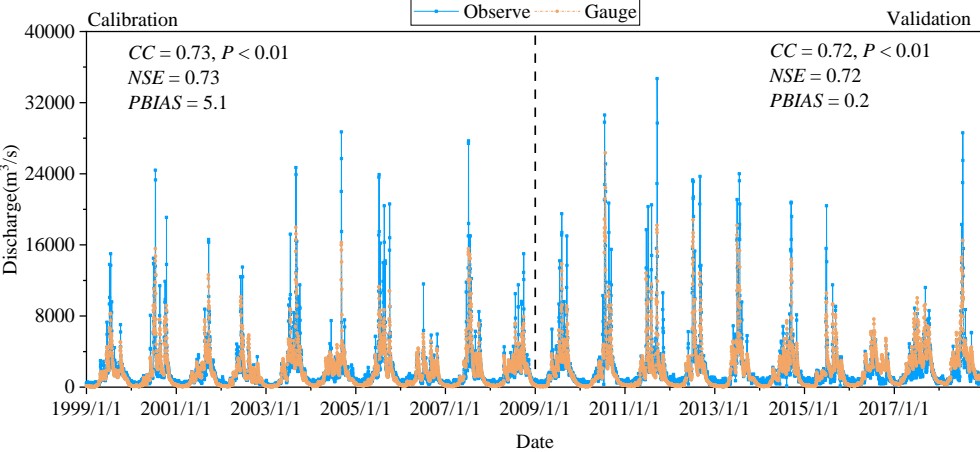


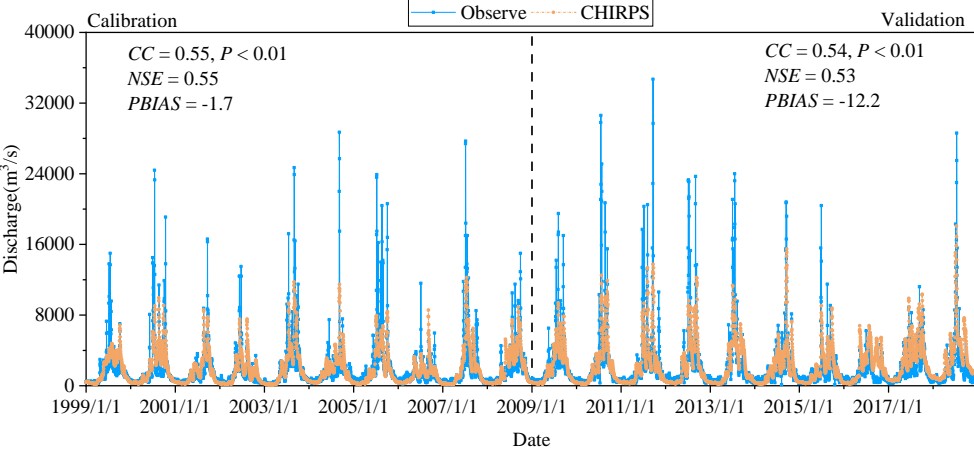

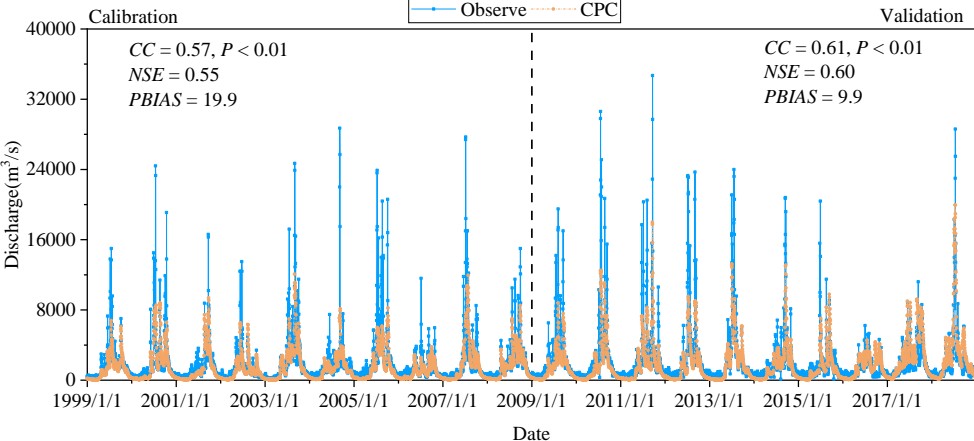

**Figure 11.** Observed and simulated discharges at the outlet of JRW at daily scale using precipitation inputs of Gauge, CHIRPS and CPC, respectively.




**Figure 12.** Observed and simulated discharges at the outlet of JRW calculating at daily scale and presenting at monthly scale using precipitation inputs of Gauge, CHIRPS and CPC, respectively.




**Figure 13.** Spatial variation of water yield at daily scale for all sub-basins calculated with precipitation inputs of (a) Gauge (b) CHIRPS and (c) CPC.





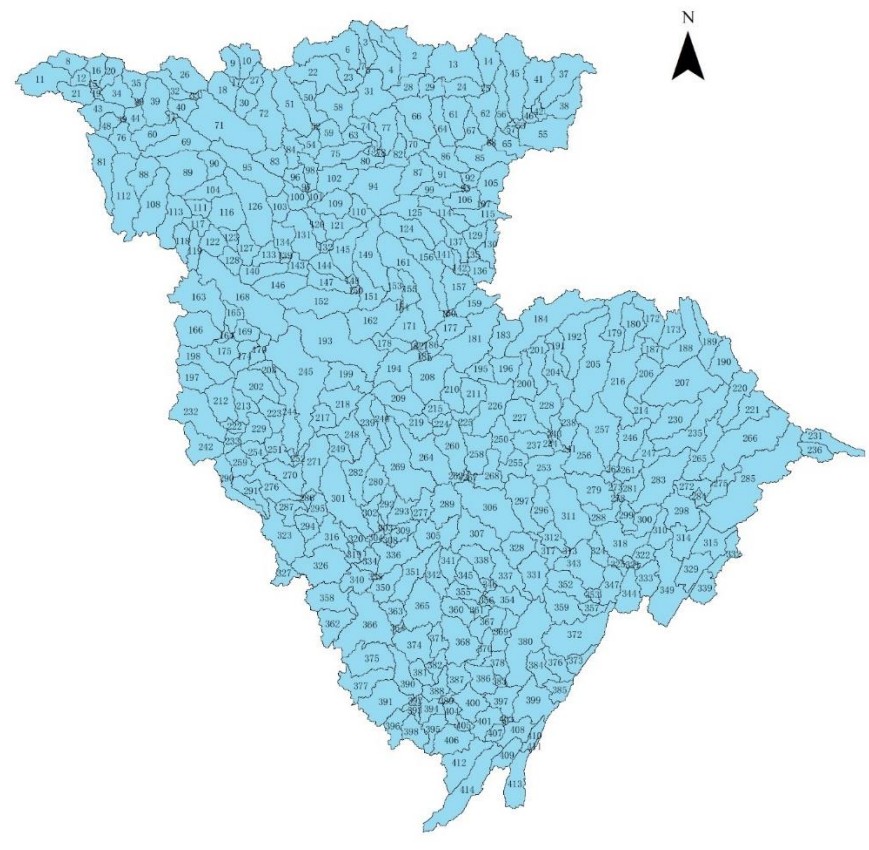

**Figure 14.** Spatial distribution of sub-basins in SWAT, named by Numbers.




**Figure 15.** Full records of flood events occurred throughout the study period and all sub-basins detected by three precipitation products, where the AR, LR, MR, HR stand for precipitation of all rainfall intensities, intensity between 0.1 and 50 mm/day, intensity between 50 and 100 mm/day, intensity more than 100 mm/day, respectively.

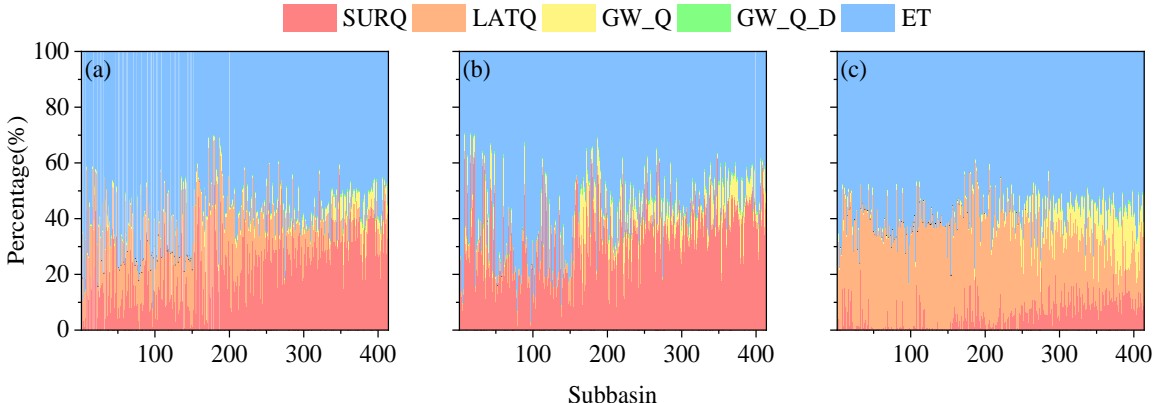


**Figure 16.** Water balance components for all sub-basins derived from SWAT models using precipitation inputs of (a) Gauge (b) CHIRPS and (CPC) (where SURQ represents surface runoff $Q_{surf}$; LATQ represents lateral flow $Q_{lat}$; GW_Q is the base flow from the shallow aquifer; GW_Q_D is the base flow from the deep aquifer, and the sum of GW_Q and GW_Q_D equals to $Q_{gw}$; ET represents evapotranspiration $E_a$.