# Peer review of "Hydrological evaluation of open-access precipitation data using SWAT at multiple temporal and spatial scales"

_Hydrology and Earth System Sciences, 2020_

## Referee Comment (RC1) · Anonymous Referee #1 · 24 Mar 2020

This paper uses two open-access precipitation products (CHIRPS and CPC) and a dataset from rain gauges to drive the SWAT hydrological model in the Jiang river watershed in China. All three precipitation datasets are shown to produce generally similar hydrological model performances, with the calibrated parameterisation reducing the effect of the identified differences in the precipitation datasets through changing hydrological processes. This is a potentially useful paper for the hydrological modelling community in that it highlights that an acceptable hydrological performance according to the commonly-used Moriasi et al (2007) criteria for the Nash Sutcliffe Efficiency metric does not mean that the hydrological processes are correctly simulated – but that it can indicate that the calibration process has merely been successful in altering the

catchment hydrological processes to compensate for inadequacies in the input data. However, I have three main concerns with the current paper: 1) The paper fails to articulate the implications of its finding (that hydrological models can give very similar model performance, with differing process behaviour, with precipitation datasets with quite different characteristics) in either the Conclusions or the Abstract. For example, Remesan and Holman (2015) study cited by the authors showed that such 'similar' calibrated/validated models, when subsequently run using perturbed inputs (e.g. climate change scenario), can lead to different magnitudes and directions of hydrological change due to their differing parameterisation. The authors should consider how their findings can guide modellers in the use of these different precipitation datasets for the hydrological modelling of the current and future climate. 2) Given that the authors are simulating a 159,000km2 catchment using a single flow gauge for calibration / validation, there is huge equifinality in their results. Given that they uised the SUFI-2 / SWATCUP, I would have expected some assessment and discussion of the uncertainty in their model results 3) The paper provides three sets of SWAT output analyses – monthly, daily and daily aggregated to monthly. However, SWAT is a daily model so the monthly SWAT outputs are themselves an internal aggregation of its daily outputs; so the presentation and description of the daily aggregated to monthly outputs (L439-448 and Figures 12 and 13) are meaningless and should be removed.

Other minor comments L19 – change "All three products" to "Both OPPs" as the text ios comparing to the gauge model L153 – is the evapotranspiration "actual", "potential" or "reference"? L169-170 – how has the classification accuracy been determined, given that it was based on "manual visual interpretation"? L194 – how does a dataset (CHIRPS v2.0) released in 2015 provide data to the "present"? L237 – looking at equation (3), isn't the optimal value of STDn = 1 e.g. identical STDs? And why should STDn values range from 0-1 which implies STD gauge can never be < STDopp? General – RMSE, STD and PBIAS have units – please use them throughout L463 – "antecedent" is the more usual term for "early-stage" L483 – there are no ALPHA-BF parameter ranges given in Table 1 and 2 to substantiate this. The values of ALHPA_BF and

GWRECH_DP should be added to the tables L486 – what is "proletarian" flow? L500
– equation 7 L560 – "streamflow photograph"? hydrograph?

---

## Referee Comment (RC2) · Anonymous Referee #2 · 20 Apr 2020

This paper presents a hydrological evaluation of two open-access precipitation products (CHIRPS and CPC) compared with rain gauge dataset, at multiple temporal and spatial scales. The content of this research is of great interest to readers of watershed hydrology, remote sensing, and satellite meteorology, since it provided valuable suggestions for researchers in these fields, especially for hydrologic modelers. It is demonstrated by the authors that, even with obvious statistical differences, performances of the three selected precipitation datasets in simulating water yield are parallel. Comparably, inconsistency were found when OPPs and rain gauge data were used to simulate hydrological components, e.g. Surface runoff, lateral flow, and base flow. Inner mechanism was highlighted from both spatial and temporal scales. Overall, this manuscript

is quite well written and presented. Minor revision comments below aim to improve the quality of the manuscript.

1. L174-182: The spatial resolutions of CHIRPS (0.05 °) and CPC (0.5 °) were higher than that of the geographic datasets, "some of the grid records are potentially missed, especially for the high - resolution CHIRPS products." Duan Z et al. (2019) proposed an area-weighted method to calculated precipitation for each subbasin, "Calculate the area-weighted average daily CHIRPS data (after disaggregated by 10 times (0.005°)) from all grids within the subbasin to represent the effective daily precipitation for each subbasin". This might be an alternative way to solve the data problem.

2. L244-248: Moriasi et al. (2007), cited by the author, used three indicators RSR (ratio of the root mean square error to the standard deviation of measured data), NSE, and PBIAS to establish a model to evaluate performance level, while the author used only two. Why not use all three metrics? Besides, since only the NSE index was graded into different evaluation levels, was the evaluation on model performance reliable without the evaluation grades from other two indicators?

3. L309-317 (Fig.8): Explain what "The correlation coefficients' spatial variation" is? The spatial correlation of the three precipitation datasets should be a value rather than a graph. Explain how Fig. 8 was calculated and obtained. Explain why distinguish average precipitation in daily and monthly scales?

Please be aware of following grammar errors and typos: 1. Double-check: L11- "Jiang River". L114- "larges" should be "largest". L299- "varations" should be "variations". 2. Grammar errors. L98- The verb "have" should be "has". L92- The article "an" here should be "a". L310- "relative" should be "relatively". 3. L342 & L360-361 unit of CC should be decimal rather than percentage. 4. L257-As IPCC reported, "Extreme rainfall" was defined as the 95th percentile of daily precipitation data. Therefore, Fig.3, shown as monthly rainfall box chart, failed to capture "extreme rainfall values" 5. L327- Usually we use "validation" instead of "verification". 6. L451- "although they performed

slightly better at the daily scale." the model should perform better at the monthly scale?

---

## Author Comment (AC1) · 19 May 2020

**Response to Comments:Reviewer#2**

We sincerely appreciate your time and comments on our manuscript. We both thank the positive remarks and the specific concerns, which provide guidance to improve the manuscript. Please see the bellowing point-to-point responses to the main concerns and minor revisions.

**General comments:** This paper presents a hydrological evaluation of two open-access precipitation products (CHIRPS and CPC) compared with rain gauge dataset, at multiple temporal and spatial scales. The content of this research is of great interest to readers of watershed hydrology, remote sensing, and satellite meteorology, since it provided valuable suggestions for researchers in these fields, especially for hydrologic modelers. It is demonstrated by the authors that, even with obvious statistical differences, performances of the three selected precipitation datasets in simulating water yield are parallel. Comparably, inconsistency were found when OPPs and rain gauge data were used to simulate hydrological components, e.g. Surface runoff, lateral flow, and base flow. Inner mechanism was highlighted from both spatial and temporal scales. Overall, this manuscript is quite well written and presented. Minor revision comments below aim to improve the quality of the manuscript.

**Authors' response:** Thank you very much for your comment and encouragement. And we greatly appreciate your suggestions which provide great suggestions to improve the manuscript.

**Specific comments:**
*1. The spatial resolutions of CHIRPS (0.05°) and CPC (0.5°) were higher than that of the geographic datasets, "some of the grid records are potentially missed, especially for the high - resolution CHIRPS products." Duan Z et al. (2019) proposed an area-weighted method to calculated precipitation for each subbasin, "Calculate the area-weighted average daily CHIRPS data (after disaggregated by 10 times (0.005°)) from all grids within the subbasin to represent the effective daily precipitation for each subbasin". This might be an alternative way to solve the data problem.*

**Authors' response:** Greatly appreciate your suggestion. Following this advice, we recalculated the precipitation inputs for each sub-basin via "area-weighted" (AW)

method, and compared the results with those derived by "Nearest Distance" (ND) adopted in this manuscript. Take the sub-basin No.411, which is located at Beibei hydrological station, as an example, the calculated precipitation inputs by the above two methods were depicted in Fig. 1. It's shown in Fig. 1 that no significant difference ($P = 0.88$) were detected by using these two methods. Compared with the ND method, the amount of rainfall obtained by AW method is slightly underestimated (*PBIAS* = -0.78%), especially when the rainfall intensity is between 50 mm and 100mm. Both methods slightly increase the uncertainty of precipitation inputs. Considering the effectiveness in producing peak discharges of streamflow in SWAT model, the ND method in the original manuscript is adopted. Actually, the "potentially missed" grid records mainly refers to CHIRPS products. More than 10000 grids within the JRW or nearby were adopted, which is much higher than that of the CPC (~165 grids) and rain gauges (20 stations). Considering that the "missed" grids could be covered by other grids within the same sub-basin due to the high resolution of CHIRPS.

**Figure 1.** plot of the CHIRPS precipitation comparing between Nearest Distance method and Area Weighted method.

Therefore, to avoid misrepresentation of data accuracy, the manuscript will be revised as follows (**line 177 to line 178**):

L177-178 – "Using this method, some of the grid records are potentially missed, especially for the high-resolution CHIRPS products." Will be revised as "Using this method, the grid records of high-resolution CHIRPS products within the same sub-basin will be uniformly assigned the grid value closest to the centroid, which will offset the high resolution advantage of CHIRPS products."

*2. Moriasi et al. (2007), cited by the author, used three indicators RSR (ratio of the root mean square error to the standard deviation of measured data), NSE, and PBIAS to establish a model to evaluate performance level, while the author used only two. Why not use all three metrics? Besides, since only the NSE index was graded into different evaluation levels, was the evaluation on model performance reliable without the evaluation grades from other two indicators?*

**Authors' response:** Greatly appreciate the comment. After an extensive literature review on model evaluation system, we followed the suggestion and adopted all three indicators to evaluate the performance of the hydrologic models with different precipitation datasets. The *RSR* indicator incorporates the benefits of error index statistics and includes a scaling/normalization factor, so this statistic indicator can be applied to various scale and object.

*RSR* value: observations standard deviation ratio (*RSR*) is an error index statistic between the OPPs and Gauge datasets. Root mean square error (*RMSE*) divided by *STD* values would derive the *RSR* value. *RSR* has a ranges from 0 to $\infty$ with 0 as the optimal value. The calculation equation is expressed as follows:

$$RSR = \frac{RMSE}{STD} = \frac{\sqrt{\sum_{i=1}^{n}(S_i - Q_i)^2}}{\sqrt{\sum_{i=1}^{n}(S_i - \overline{S})^2}}, \tag{4}$$

The models' performances were classified using *RSR*, *NSE*, and percentage bias (*PBIAS*) values defined by Moriasi et al. (2007), which are shown in Table 2. And the evaluation results by using all three metrics at monthly and daily scales were depicted in Table S1 and Table S2, respectively.
Comparably, in the original manuscript, like most existing papers did (Zhu et al., 2015; Tuo et al., 2018; Duan et al., 2019), only *NSE*, *CC*, and *PBIAS* were adopted to evaluate the simulation performance of hydrologic models. But they did not give a classified evaluation of the simulation results. When we used only *NSE* to classify the simulation performance of the model, as presented in the original manuscript, the evaluation results are "Very good" for all three models at the monthly scale, but when we used three indicators to evaluate, CPC only reached the level of "Good" (Table S1.).

**Table 2.** General performance ratings statistics recommended by Moriasi et al. (2007).

| Performance Rating | *RSR* | *NSE* | *PBIAS* |
|---|---|---|---|
| Very good | $0.00 < RSR \leq 0.50$ | $0.75 < NSE \leq 1.00$ | $PBIAS \leq \pm 10$ |
| Good | $0.50 < RSR \leq 0.60$ | $0.65 < NSE \leq 0.75$ | $\pm 10 < PBIAS \leq \pm 15$ |
| Satisfactory | $0.60 < RSR \leq 0.70$ | $0.50 < NSE \leq 0.65$ | $\pm 15 < PBIAS \leq \pm 25$ |
| Unsatisfactory | $RSR > 0.70$ | $NSE < 0.50$ | $PBIAS > \pm 25$ |

**Table S1.** Evaluation results of monthly scale SWAT model

| | | Calibration | | | Validation | |
|---|---|---|---|---|---|---|
| | Gauge | CHIRPS | CPC | Gauge | CHIRPS | CPC |
| *RSR* | 0.28 | 0.42 | 0.36 | 0.36 | 0.49 | 0.46 |
| *NSE* | 0.92 | 0.82 | 0.87 | 0.87 | 0.76 | 0.79 |
| *PBIAS* (%) | 7.9 | 2.3 | 10.8 | 1.2 | -6.6 | 4.2 |
| Evaluation | Very good | Very good | Good | Very good | Very good | Very good |

**Table S2.** Evaluation results of daily scale SWAT model

| | | Calibration | | | Validation | |
|---|---|---|---|---|---|---|
| | Gauge | CHIRPS | CPC | Gauge | CHIRPS | CPC |
| *RSR* | 0.52 | 0.67 | 0.67 | 0.53 | 0.69 | 0.63 |
| *NSE* | 0.73 | 0.55 | 0.55 | 0.72 | 0.53 | 0.6 |
| *PBIAS* (%) | 5.1 | -1.7 | 19.9 | 0.2 | -12.2 | 9.9 |
| Evaluation | Good | Satisfactory | Satisfactory | Good | Satisfactory | Satisfactory |

Duan, Z., Tuo, Y., Liu, J., Gao, H., Song, X., Zhang, Z., Yang, L., and Mekonnen, D. F.: Hydrological evaluation of open-access precipitation and air temperature datasets using SWAT in a poorly gauged basin in Ethiopia, J. Hydrol., 569, 612-626, https://doi.org/10.1016/j.jhydrol.2018.12.026, 2019.

Tuo, Y., Marcolini, G., Disse, M., and Chiogna, G.: A multi-objective approach to improve SWAT model calibration in alpine catchments, J. Hydrol., 559, 347-360, https://doi.org/10.1016/j.jhydrol.2018.02.055, 2018.

Zhu, H., Li, Y., Liu, Z., Shi, X., Fu, B., and Xing, Z.: Using SWAT to simulate streamflow in Huifa River basin with ground and Fengyun precipitation data, J. Hydroinform., 17, 834–844, https://doi.org/10.2166/hydro.2015.104, 2015.

[Figure]

*3. Explain what "The correlation coefficients' spatial variation" is? The spatial correlation of the three precipitation datasets should be a value rather than a graph. Explain how Fig. 8 was calculated and obtained. Explain why distinguish average precipitation in daily and monthly scales?*

**Authors' Response:** Thanks a lot for the question. Basically, the term "correlation coefficients" ($CC_{sub}$) refers to the correlation between two OPPs and Gauge at either daily or monthly scales for each sub-basin. "The correlation coefficients' spatial variation" is the variation of $CC_{sub}$ in different sub-basins. The calculation steps of $CC_{sub}$ are as follows. Firstly, the spatial distribution diagrams of Gauge records and the two OPPs are calculated. The correlation coefficient for each sub-basin between the Gauge series and OPPs series is calculated by using the correlation coefficient method. Coefficients for the all 414 sub-basins will form a spatial distribution plot of the correlation coefficients. Fig. 8 aims to distinguish the correlation of rainfall amounts in different sub-basins, and to preliminarily judge the performance of OPPs in different sub-basins, especially at different time scales. Distinguish average precipitation at daily and monthly scales are important, since the difference in precipitation amount statistics may highly resulted in different modelling performance and inner mechanics of hydrologic processes. Actually, the *CC* values between spatial-aggregated CPC and Gauge records are higher than that of CHIRPS and Gauge at both scales, yet the *CC* values at the monthly scale are much higher than that at daily scale. Similar variation regularity was found in spatial distribution, which was described in Fig. 8. However, *CC* values between CPC and Gauge are smaller than that of CHIRPS and Gauge at a portion of sub-basins, which was explained at **line313** to **line315** in manuscript:
*"Spatially, the higher CC values between the Gauge and CPC at the monthly scale are mainly distributed in areas with comparably low or high rainfall amounts, such as Wudu and Wangyuan. Yet, the CC value was less relevant in areas with moderate rainfall (e.g., Suining), compared with to that of the Gauge and CHIRPS."*

The manuscript will be accordingly revised as follows ( **line313** to **line315** ):

**Original version:**
L311-315 – "The correlation coefficients' spatial variation between the Gauge and OPPs at monthly and daily scales are illustrated in Fig. 8. Overall, the monthly scale *CC* values (with a rang of 0.7 1) are comparably larger than that of the daily scale values (with a rang of 0.5 0.7). Spatially, the higher *CC* values between the Gauge and CPC at the monthly scale are mainly distributed in areas with comparably low or high rainfall amounts, such as Wudu and Wangyuan. Yet, the *CC* value was less relevant in areas with moderate rainfall (e.g., Suining) relative to that of the Gauge and CHIRPS."

**Revised version:**
"The *CC*, $STD_n$, and *RSR* values of precipitation spatial distribution between CHIRPS and Gauge are 0.89, 0.96, and 0.55, respectively, and 0.82, 0.87, 0.62 between CPC and Gauge, respectively. These statistics indicate that both CHIRPS and CPC estimates can describe the spatial distribution of precipitation in JRW, among which CHIRPS depicts better performance. The correlation coefficients between the Gauge and OPPs at monthly or daily scales for every sub-basin are illustrated in Fig. 8. Overall, the *CC* values at monthly scale (with a rang of 0.7 1) are comparably larger than that of the daily scale (with a rang of 0.5 0.7). Spatially, the higher *CC* values between the Gauge and CPC at the monthly scale are mainly distributed in areas with comparably low or high rainfall amounts, such as Wudu and Wangyuan. Yet, the *CC* value was less relevant in areas with moderate rainfall (e.g., Suining), compared with to that of the Gauge and CHIRPS.

**Minor revisions:**

1. Double-check: L11- "Jiang River". L114- "larges" should be "largest". L299-"varations" should be "variations".

 **Authors' Response:** Thanks a lot for pointing out the typos, and we will make revisions all through the revised manuscript.

 L11 – "the **Jiang** River Watershed (JRW)" will be corrected as "the **Jialing River** Watershed (JRW)"

 L114 – "the **larges** drainage area" will be corrected as "the **largest** drainage area"

 L299 – "Spatial **varations**" will be corrected as "Spatial **variations**"

Other typos in the paper will be also revised, which are listed as below:

 L214 – "**base flow**" will be revised as "**baseflow**"

 L218 – "**by pass**" will be revised as "**bypass**"
2. Grammar errors. L98- The verb "have" should be "has". L92- The article "an" here should be "a". L310- "relative" should be "relatively".

**Authors' Response:** Thank you very much for pointing out the grammar errors, and we will correct them as follows:

L98 – "**have** not been fully investigated." will be corrected as "**has** not been fully investigated."

L92 – "Bai & Liu (2018) used **an** HIMS model ..." will be corrected as "Bai & Liu (2018) used **a** HIMS model..."

L310 – "the overall precipitation values estimated by the CHIRPS are **relative** higher" will be corrected as "the overall precipitation values estimated by the CHIRPS are **relatively** higher"

3. L342 & L360-361 unit of CC should be decimal rather than percentage.

**Authors' Response:** Thank you so much for the suggestion.

L342 – "The spatial correlation between WYLD and precipitation for rainfall for the Gauge, CHIRPS, and CPC products reached **84.8 %, 84.3 %, and 90.84 %**, respectively." will be corrected as "The spatial correlation between WYLD and precipitation for rainfall for the Gauge, CHIRPS, and CPC products reached **0.85, 0.84, and 0.91**, respectively."

[Figure]

L360-361 – "The it CC values between the WYLD and precipitation for Gauge, CHIRPS, and CPC at the daily scale are **83.45 %, 84.41 % and 91.70 %**, respectively, . . ." will be corrected as "The *CC* values between the WYLD and precipitation for Gauge, CHIRPS, and CPC at the daily scale are **0.83, 0.84 and 0.92**, respectively, . . ."

4. L257-As IPCC reported, "Extreme rainfall" was defined as the 95th percentile of daily precipitation data. Therefore, Fig.3, shown as monthly rainfall box chart, failed to capture "extreme rainfall values"

**Authors' Response:** Thanks a lot for the comment and suggestion.

The term "extreme rainfall" here means that estimation of Gauge monthly rainfall in the rainy season (especially in July) is significantly higher than that of the other two OPPs. For better interpretation, the term will be revised as:

L257 – "Fig. 3 shows that **the extreme rainfall values** captured by Gauge are higher than those of CHIRPS and CPC." will be revised as"Fig. 3 shows that **the rainfall values in the rainy season (especially in July)** captured by Gauge are higher than those of CHIRPS and CPC."

5. L327-Usually we use "validation" instead of "verification".

**Authors' Response:** Thank you very much for your advice, and we will revise this term into validation all through the manuscript:

L327 – "Moriasi et al. (2007), all three models, each using a different precipitation product, achieved "very good" performance for both the calibration and **verification** periods, ..." will be corrected as "Moriasi et al. (2007), all three models, each using a different precipitation product, achieved "very good" performance for both the calibration and **validation** periods, ..."

L336 – "CPC showed significant overestimation in 2017 and 2018 during the **verification** period." will be corrected as "CPC showed significant overestimation in 2017 and 2018 during the **validation** period."

6. L451- "although they performed slightly better at the daily scale." the model should perform better at the monthly scale?

**Authors' Response:** We apologize for this error, and we will revise it as follows:

L451 – "although they performed **slightly better** at the daily scale." will be corrected as "although they performed **slightly worse** at the daily scale."

[Figure]

**Fig. 1.** Scatter plot of the CHIRPS precipitation comparing between Nearest Distance method and Area Weighted method.

---

## Author Comment (AC2) · 26 May 2020

**Response to Referee Comments:Reviewer#1**

First of all, we sincerely appreciate the comments on our manuscript. We both thank the positive remarks and the specific concerns, which provided great encouragement and specific guidance to the authors to improve the manuscript. Please see the bellowing point-to-point responses to the main concerns and minor comments.

**General comments:**

*This paper presents a hydrological evaluation of two open-access precipitation products (CHIRPS and CPC) compared with rain gauge dataset, at multiple temporal and spatial scales. The content of this research is of great interest to readers of watershed hydrology, remote sensing, and satellite meteorology, since it provided valuable suggestions for researchers in these fields, especially for hydrologic modelers. It is demonstrated by the authors that, even with obvious statistical differences, performances of the three selected precipitation datasets in simulating water yield are parallel. Comparably, inconsistency were found when OPPs and rain gauge data were used to simulate hydrological components, e.g. Surface runoff, lateral flow, and base flow. Inner mechanism was highlighted from both spatial and temporal scales. Overall, this manuscript is quite well written and presented. Minor revision comments below aim to improve the quality of the manuscript.*

**Main concerns:**

*1. The paper fails to articulate the implications of its finding (that hydrological models can give very similar model performance, with differing process behaviour, with precipitation datasets with quite different characteristics) in either the Conclusions or the Abstract. For example, Remesan and Holman (2015) study cited by the authors showed that such 'similar' calibrated/validated models, when subsequently run using perturbed inputs (e.g. climate change scenario), can lead to different magnitudes and directions of hydrological change due to their differing parameterization. The authors should consider how their findings can guide modelers in the use of these different precipitation datasets for the hydrological modelling of the current and future climate.*

**Authors' response:** Greatly appreciate the comment and suggestion. As stated in Remesan and Holman's (2015) study, "with similar historical model performance, model construction with different baseline meteorological data choices significantly

condition the magnitude and direction of simulated hydrological impacts of climate change", the current study has reached "similar" conclusions: "with similar performances in simulating river runoff, different types of precipitation data digested in hydrologic modeling tends to counterbalance their identified differences by differing parameterization and leads to different directions of hydrologic processes". Considering that this research focuses on precipitation condition under current climate, it could generally provide implications to hydrological modelers of current and future climate from following two aspects:

1) From perspective of precipitation estimation: CHIRPS has a higher spatial resolution (with 0.05° being equivalent to a resolution of one gauge station for every 30.25 km$^2$ area) and a stronger ability to recognize heavy rain and extreme rainfall (Fig.4 – Fig.6 in the manuscript). These features would facilitate the widespread use of CHIRPS in future climate analyses. Take extreme climate analyses for example, it is reported that the frequency of extreme rainfall events in China has been significantly increased in past decades and this tendency will continue increasing in future climate change (Mou et al., 2020; Xi et al., 2018). With this background in future climate change, CHIRPS would provide high potential in future extreme rainfall event analyses with high spatial resolution. Actually, CHIRPS has been applied to identify extreme rainfall events by indicators of nP (Number of days with P ≥1mm), PRCPTOT (Annual total precipitation), and R95pad (Total precipitation when P >95 percentile of all days), etc. (Cavalcante et al., 2020). In contrast, the CPC's strong ability to identify light rain represents a unique advantage in extreme drought-related research.

2) From perspective of hydrologic modeling: overall, the three precipitation types derive almost equivalent and acceptable hydrological performance according to Moriasi et al's criteria (2007), while CHIRPS presented better performance in uncertainty analyses. Although the river runoff values simulated by the three models are basically

consistent, there are significant differences among other hydrological components, such as surface runoff, lateral flow, and base flow. CHIRPS tends to derive more surface flow due to the higher precipitation detection, while CPC tends to yield more lateral flow due to the lower precipitation detection. As such, CHIRPS would suit broader applications in flood prediction of the future climate due to its ability in extreme precipitation identification and surface flow simulation. More importantly, multiple-objective calibration based on multiple hydrological components are recommended to improve SWAT modeling in large and spatial resolved watershed.

Cavalcante, R. B. L., Ferreira, D. B. da S., Pontes, P. R. M., Tedeschi, R. G., da Costa, C. P. W., & de Souza, E. B. (2020). Evaluation of extreme rainfall indices from CHIRPS precipitation estimates over the Brazilian Amazonia. Atmos. Res., 238, 104879. doi:10.1016/j.atmosres.2020.104879.

Mou, S., Shi, P., Qu, S., Feng, Y., Chen, C., & Dong, F. (2020). Projected regional responses of precipitation extremes and their joint probabilistic behaviors to climate change in the upper and middle reaches of Huaihe River Basin, China. Atmos. Res., 104942. doi:10.1016/j.atmosres.2020.104942.

Xi, Y., Miao, C., Wu, J., Duan, Q., Lei, X., & Li, H. (2018). Spatiotemporal Changes in Extreme Temperature and Precipitation Events in the Three-Rivers Headwater Region, China. J. Geophys. Res-Atmos., 123, 5827–5844. doi:10.1029/2017jd028226.

The above considerations will be articulated in sections of **Abstract**, **Discussion** and **Conclusions** of the revised manuscript.

**Original version of abstract:**

[revised manuscript text omitted]

*2. Given that the authors are simulating a 159,000km² catchment using a single flow gauge for calibration / validation, there is huge equifinality in their results. Given that they used the SUFI-2 / SWATCUP, I would have expected some assessment and discussion of the uncertainty in their model results.*

**Authors' response:** Greatly appreciate the comment. Assessment and discussions on the uncertainty of model results are quite important issues in hydrologic modelling (Abbaspour, 2015). In our study, the model calibration / validation use a single hydrologic station, with a monitored area of more than 159000 km², which would induced inevitable system or random deviation by parameter calibration. Therefore, as the comment suggested, uncertainty analyses on model results should be processed and discussed.

Abbaspour, K. C. (2015) SWAT-CUP 2012: SWAT Calibration and Uncertainty Programs - A User Manual. Tech. rep., Swiss Federal In-stitute of Aquatic Science and Technology, Eawag, Dübendorf, Switzerland.

With the considerations above, assessment and discussion on the uncertainty of model results will be added in the revised manuscript, and the modification will be specified as follows:

**Original version of abstract:**
 L17-18 – "**All three products** satisfactorily reproduce the stream discharges at the JRW outlet with better performance than the Gauge model."

**Revised version of abstract:**
 **Both OPPs** satisfactorily reproduce the stream discharges at the JRW outlet with slightly worse performance than the Gauge model. Model with CHIRPS as inputs

performed slightly better in both model simulation and uncertainty analysis than that of CPC.

**Revised version of methodology section:**

*At the end of Sect.2.4.2, we added a description of the SWAT-CUP-based uncertainty analysis method:*

[revised manuscript text omitted]

*3. The paper provides three sets of SWAT output analyses – monthly, daily and daily aggregated to monthly. However, SWAT is a daily model so the monthly SWAT outputs are themselves an internal aggregation of its daily outputs; so the presentation and description of the daily aggregated to monthly outputs (L439-448 and Figures 12 and 13) are meaningless and should be removed.*

**Authors' Response:** Thanks a lot for this comment and advice. The presentation and description of the daily aggregated to monthly outputs (L439-448 and Figures 12 and 13) will be removed in the revised manuscript.

As one of the major objectives of this manuscript was to evaluate the performances of different precipitation datasets in simulating the watershed streamflow using SWAT on different temporal scales, the authors ran the SWAT models at monthly and daily scales, respectively. Essentially, SWAT is a daily model that monthly outputs can be derived by aggregating its daily outputs. For researchers, who are not able to collect daily streamflow records, may be more interested in the performance at monthly scale. With this consideration, the authors presented two sets of SWAT output analyses, i.e. daily and monthly, and further look into the corresponding water balance components (Fig.4 & Table 5) adjusting by calibrated parameters (Table 2). In the previous manuscript, proportions of water balance components at monthly scale were compared and analyzed. In the revised manuscript, water balance components calculated at daily scale should also be presented and compared with results of monthly-scaled models.

**Figure 4.** Water balance components for all sub-basins derived from SWAT models using precipitation inputs of (a) Gauge (b) CHIRPS and (c) CPC at monthly scale and (d) Gauge (e) CHIRPS and (f) CPC at daily scale (where SURQ represents surface runoff $Q_{surf}$; LATQ represents lateral flow $Q_{lat}$; GW_Q is the baseflow from the shallow aquifer; GW_Q_D is the baseflow from the deep aquifer, and the sum of GW_Q and GW_Q_D equals to $Q_{gw}$; ET represents actual evapotranspiration *ET*. (Fig.16 in the manuscript)

**Table 2: Optimal parameters calibrated for all three models. (excerpts)**

| Parameters | Initial range | Gauge Monthly | Daily | CHIRPS Monthly | Daily | CPC Monthly | Daily |
|---|---|---|---|---|---|---|---|
| a__SOL_K().sol | −10/10 | 1.988/10 | -0.706/10 | -0.471/7.681 | -0.396/10 | 5.264/10 | -2.106/10 |
| v__ESCO.hru | 0/1 | 0.879/1 | 0.405/1 | 0.775/1 | 0.355/1 | 0.914/1 | 0.462/1 |
| v__ALPHA_BF.gw | 0/1 | 0.401/0.963 | 0.299/0.896 | 0.055/0.677 | 0.183/0.728 | 0.216/0.901 | 0.415/1 |

**Table 5: Summarization of annual average water balance components of the three models for the whole JRW.**

| Time scale | Datasets | Statistics | SURQ | LATQ | GW_Q | GW_Q_D | ET | Summation |
|---|---|---|---|---|---|---|---|---|
| Monthly | Gauge | Average amount/mm | 4500.00 | 2977.22 | 299.07 | 60.61 | 9076.60 | 16913.50 |
| | | Percentage/% | 26.61% | 17.60% | 1.77% | 0.36% | 53.66% | |
| | CHIRPS | Average amount/mm | 6068.35 | 773.24 | 949.56 | 140.79 | 9046.83 | 16978.78 |
| | | Percentage/% | 35.74% | 4.55% | 5.59% | 0.83% | 53.28% | |
| | CPC | Average amount/mm | 1087.19 | 5577.20 | 583.45 | 30.15 | 8694.40 | 15972.40 |
| | | Percentage/% | 6.81% | 34.92% | 3.65% | 0.19% | 54.43% | |
| Daily | Gauge | Average amount/mm | 5544.88 | 1856.00 | 244.94 | 48.29 | 9309.37 | 17003.48 |
| | | Percentage/% | 32.61% | 10.92% | 1.44% | 0.28% | 54.75% | |
| | CHIRPS | Average amount/mm | 6202.63 | 834.78 | 1167.37 | 59.75 | 10434.58 | 18699.11 |
| | | Percentage/% | 33.17% | 4.46% | 6.24% | 0.32% | 55.80% | |
| | CPC | Average amount/mm | 2493.11 | 2302.28 | 1709.95 | 88.66 | 9384.90 | 15978.90 |
| | | Percentage/% | 15.60% | 14.41% | 10.70% | 0.55% | 58.73% | |

**Results showed that:**

(1) Either at daily scale or monthly scale, all three models achieved acceptable and similar simulation performance for comparisons of both time series and spatial distributions. However, the parameter systems are completely different at two temporal scales (Table 2). The non-uniqueness of parameters has been proved a persistent drawback of SWAT (Abbaspour et al., 2004; Abbaspour, 2015; Zhang et al., 2015). And we had explained this drawback at **line399** to **line404** in manuscript:

*"In general, simulated and observed streamflow hydrographs, using OPPs and Gauge inputs, can successfully match at both monthly and daily scales. However, consistency between simulated and observed streamflow does not guarantee identical hydrologic processes. For example, the SWAT model calibrated parameters are not the same for all precipitation inputs, meaning that the hydrologic mechanics during SWAT modelling are also different. As such, it is critical that researchers and decision makers adequately understand the benefits and limitations of different precipitation products in modelling the hydrologic processes."*

(2) With differing parameterizations, different precipitation inputs tend to derive completely different hydrological component amounts at different time scales (Fig. 16 & Table 5).

Abbaspour, K. C., Johnson, C. A., & van Genuchten, M. T. (2004). Estimating Uncertain Flow and Transport Parameters Using a Sequential Uncertainty Fitting Procedure. Vadose Zone Journal, 3(4), 1340. doi:10.2136/vzj2004.1340.

Abbaspour, K. C. (2015) SWAT-CUP 2012: SWAT Calibration and Uncertainty Programs - A User Manual. Tech. rep., Swiss Federal In-stitute of Aquatic Science and Technology, Eawag, Dübendorf, Switzerland.

Zhang, J., Li, Q., Guo, B., & Gong, H. (2015). The comparative study of multi-site uncertainty evaluation method based on SWAT model. Hydrological Processes, 29(13), 2994–3009. doi:10.1002/hyp.10380.

With the considerations above, discussion on the model parameters and water balance components will be added in the revised manuscript, and the modification will be specified as follows:

**Original version of conclusions section:**

[revised manuscript text omitted]

**Minor revision comments:**

1. L19 – change "All three products" to "Both OPPs" as the text is comparing to the gauge model.

**Authors' Response:** Thanks a lot for pointing out this issue.

The sentence will be corrected as "Both OPPs satisfactorily reproduce the stream discharges at the JRW outlet with slightly worse performance than the Gauge model, . . ."

2. L153 – is the evapotranspiration "actual", "potential" or "reference"?

**Authors' Response:** It's the actual evapotranspiration, and the sentence will be revised as "the **annual average actual evapotranspiration (*ET*)** ranges from 800 to 1000 mm."
The descriptions related to evapotranspiration all through the manuscript have been corrected in the revised manuscript:

L214 – "Water balance, including precipitation, surface runoff, **evapotranspiration**, infiltration, lateral and base flow, and percolation to shallow and deep aquifers, is mathematically expressed as follows:"
The sentence will be corrected as "Water balance, including precipitation, surface runoff, **actual evapotranspiration**, infiltration, lateral and base flow, and percolation to shallow and deep aquifers, is mathematically expressed as follows:"

L217 – "$E_a$ = evapotranspiration" will be corrected as "$ET$ = actual evapotranspiration".

L410-411 – "However, they do share some similarities in that the **evapotranspiration ($ET$)** portions of all three products are above 50 %, resulting in a watershed runoff production coefficient of 0̃.45." The sentence will be corrected as "However, they do share some similarities in that the **actual evapotranspiration ($ET$)** portions of all three products are above 50 %, resulting in a watershed runoff production coefficient of 0̃.45."

L418-422 – "The SWAT model indirectly increases WYLD by using higher ESCO and thus decreases the $ET$ value. In this study, the ESCO values for Gauge, CHIRPS, and CPC range from 0.879 - 1, 0.775 – 1, and 0.914 - 1, respectively. Furthermore, the total $ET$ values during the study period were 8153.94, 8161.22, and 7806.84 mm, respectively. Apparently, the CPC model reduced its corresponding $ET$ by using a higher ESCO parameter, so that the lack of precipitation inputs would be offset by less evaporation."
The sentence will be corrected as "The SWAT model indirectly increases WYLD by using higher ESCO and thus decreases the $ET$ value. In this study, the ESCO values for Gauge, CHIRPS, and CPC range from 0.879 - 1, 0.775 – 1, and 0.914 - 1, respectively. Furthermore, the total $ET$ values during the study period were 8153.94, 8161.22, and 7806.84 mm, respectively. Apparently, the CPC model reduced its corresponding $ET$ by using a higher ESCO parameter, so that the lack of precipitation inputs would be offset by less $ET$."

3. L169-170 – how has the classification accuracy been determined, given that it was based on "manual visual interpretation"?

**Authors' Response: Thank you very much for this question.**

The procedure of deriving LUCC types based on 2010 Landsat TM/ETM remote sensing images are as follows: The geometric shape, colour feature, texture feature and spatial distribution of ground objects were analysed and extracted according to the image spectral features. The remote sensing image interpretation marks were established based on the field measurement data and the reference map. Six primary classifications were recognized- cultivated land, woodland, grassland, water area, construction land, and unused land. The quality of the LUCC product was checked by combining field survey and random sampling dynamic map spot for repeated interpretation analysis. Generally, the quality inspection result is that the classification accuracy of cultivated land data is ẽ85%, and that of other data can reach more than 75%.

The manuscript will be revised as "The data included six primary classification-s—cultivated land, woodland, grassland, water area, construction land, and unused land, as well as 25 secondary classifications. After checking the quality of data products by combining field survey and random sampling dynamic map spot for repeated interpretation analysis, it is proved that the cultivated land's classification accuracy was 85 %, and other data classification accuracies reached 75 %.".

4. L194 – how does a dataset (CHIRPS v2.0) released in 2015 provide data to the "present"?

**Authors' Response: Thank you very much for this question.**
Actually, the CHIRPS v2.0 dataset has been continuously updated since it was released in 2015, and we are sorry for the misinterpretation.
The manuscript will be revised as "The first gridded format CHIRPS product was released in 1981 to present and the most recent one (V2.0 datasets) was released in February 2015. The dataset spans from 1981 to the present and provides daily precipitation data with a spatial resolution of 0.05° in a pseudo global coverage of 50° N - 50° S."

5. L237 – looking at equation (3), isn't the optimal value of $STD_n$ = 1 e.g. identical $STD_s$? And why should STDn values range from 0-1 which implies $STD$ gauge can never be < $STD$ opp? General –$RMSE$, $STD$ and $PBIAS$ have units – please use them throughout

**Authors' Response:** we are sorry to make this mistake for our neglect, which should be corrected as: "The $STD_n$ values range from 0 to $\infty$, and the optimal value is 1."
The units of $RMSE$, $STD$ and $PBIAS$ will be revised throughout the manuscript as follows:

L189-190 – "Where *n* is the number of the time series; $Q_i$ and $S_i$ are measured values and estimated values (or simulated values), respectively; and $\overline{Q}$ and $\overline{s}$ are the mean values of the measured and estimated values (or simulated values), respectively."
The sentence will be revised as "Where *n* is the number of the time series; $Q_i$ and $S_i$ are measured values and estimated values, respectively; and $\overline{Q}$ and $\overline{s}$ are the mean values of the measured and estimated values, respectively. **The value may refer to either precipitation (mm) or streamflow discharge (m$^3$/s)**."

[Figure]

L191 – "Standard deviation (*STD*) represents the discretization degree of the datasets." The sentence will be revised as "Standard deviation (*STD*) represents the discretization degree of the precipitation datasets (**mm**)."

L195-196 – "*RMSD* value: Root mean square deviation (*RMSD*) is used to demonstrate the error between the OPPs and Gauge datasets. *RMSD* has a range from 0 to $+\infty$, and an optimal value of 0." The sentence will be revised as "*RMSD* value: Root mean square deviation (*RMSD*) is used to demonstrate the error between the OPPs and Gauge datasets (**mm**). *RMSD* has a range from 0 to $+\infty$ **mm**, with an optimal value of 0 **mm**."

L248 – "*PBIAS* describes the OPPs' systematic bias. *PBIAS* ranges from 0 to $+\infty$, and the optimal value is 0." The sentence will be revised as "*PBIAS* describes the OPPs' systematic bias (**%**). *PBIAS* ranges from 0 to $+\infty$ **%**, and the optimal value is 0 **%**."

L266-267 – "The *RMSD* values for Gauge-CHIRPS and Gauge-CPC are 15.80 and 12.95, respectively." The sentence will be revised as "The *RMSD* values for Gauge-CHIRPS and Gauge-CPC are 15.80 **mm** and 12.95 **mm**, respectively."

L281-283 – "Statistically, the *CC*, *STD*, and *RMSD* values between CHIRPS and the Gauge records are 0.53, 1.14, and 5.16, respectively, and 0.64, 0.87, and 3.95, respectively, between the CPC and Gauge products."
The sentence will be revised as "Statistically, the *CC*, $STD_n$, and *RMSD* values between CHIRPS and the Gauge records are 0.53 **mm** , 1.14 **mm** , and 5.16 **mm** , respectively, and 0.64 **mm** , 0.87 **mm** , and 3.95 **mm** , respectively, between the CPC and Gauge products."

L361-362 – "The *CC*, $STD_n$, and *RMSD* values between CHIRPS and Gauge are 0.92, 1.06, and 0.23, respectively, and 0.81, 0.94, 0.33 between CPC and Gauge, respectively." The sentence will be revised as "The *CC*, $STD_n$, and *RMSD* values between CHIRPS and Gauge are 0.92 **mm**, 1.06 **mm**, and 0.23 **mm**, respectively, and 0.81 **mm**, 0.94 **mm**, 0.33 **mm** between CPC and Gauge, respectively."

L266 – "The **STD** values for Gauge-CHIRPS and Gauge-CPC are 1.06 and 0.94, respectively." The sentence will be revised as "The **$STD_n$** values for Gauge-CHIRPS and Gauge-CPC are 1.06 and 0.94, respectively."

L268-270 – "Nevertheless, *PBIAS* values of Gauge-CHIRPS and Gauge-CPC were 9.58 and -6.70, respectively"
The sentence will be revised as "Nevertheless, *PBIAS* values of Gauge-CHIRPS and Gauge-CPC were 9.58 **%** and -6.70 **%**, respectively"

L343-344 – "The *PBIAS* values for Gauge-CHIRPS and Gauge-CPC are 5.85 and -5.38, respectively." The sentence will be revised as "The *PBIAS* values for Gauge-CHIRPS and Gauge-CPC are 5.85 **%** and -5.38 **%**, respectively."

6. L463 – "antecedent" is the more usual term for "early-stage".

**Authors' Response: Thank you very much for your advice, and we have revised this term into antecedent all through the manuscript:**

L373-375 – "Solano-Rivera et al. (2019) experimented in the San Lorencito headwater catchment and found that the rainfall-runoff dynamics before extreme events were mainly related to **early-stage** conditions. After extreme flood events, **early-stage** conditions had no effect on rainfall-runoff processes, and rainfall significantly affected the streamflow discharge."

The sentence will be changed as "Solano-Rivera et al. (2019) experimented in the San Lorencito headwater catchment and found that the rainfall-runoff dynamics before extreme events were mainly related to **antecedent** conditions. After extreme flood events, **antecedent** conditions had no effect on rainfall-runoff processes, and rainfall significantly affected the streamflow discharge."

7. L483 – there are no ALPHA-BF parameter ranges given in Table 1 and 2 to substantiate this. The values of ALHPA_BF and GWRECH_DP should be added to the tables.

**Authors' Response:** Thanks a lot for pointing out this error, and ALHPA_BF has been added to Table1 and Table2. Since the parameter RCHRG_DP is not a sensitive one, so it was not included in the calibration process.

**Table 1: Hydrological parameters considered for sensitivity analysis ("a_", "v_" and r_" means an absolute increase, a replacement, and a relative change to the initial parameter values, respectively).**

| Parameters | Description | Range | Default |
|---|---|---|---|
| v__ PLAPS.sub | Precipitation lapse rate[mm] | -1000/1000 | 0 |
| **v__ALPHA_BF.gw** | **Baseflow alpha factor [days$^{-1}$]** | **0/1** | **0.048** |
| v__ALPHA_BNK.rte | Baseflow alpha factor for bank storage | 0/1 | 0 |

**Table 2: Optimal parameters calibrated for all three models. (excerpts)**

| Parameters | Initial range | Gauge | | CHIRPS | | CPC | |
|---|---|---|---|---|---|---|---|
| | | Monthly | Daily | Monthly | Daily | Monthly | Daily |
| v__PLAPS.sub | −1000/1000 | 0.012/0.067 | 0.061/0.183 | 0.079/0.135 | 0.068/0.205 | 0.017/0.078 | -0.014/0.095 |
| v__ALPHA_BF.gw | 0/1 | 0.401/0.963 | 0.299/0.896 | 0.055/0.677 | 0.183/0.728 | 0.216/0.901 | 0.415/1 |
| v__ALPHA_BNK.rte | 0/1 | 0.492/0.863 | 0.444/1 | 0.201/0.696 | 0.467/1 | 0.564/1 | 0.307/0.92 |

8. L486 – what is "proletarian" flow?

 **Authors' Response:** Thank you so much for pointing out this typo.
The authors tended to articulate that "For CPC dataset, the high proportion of LR events will lead to severe rainfall losses in the initial- and post- loss processes, resulting in very limited surface water yield. As such, the sentence will be corrected as:
"A potential reason for this phenomenon may be that the rainfall during LR events tends to be easily lost in the initial- and post-loss processes, resulting in very limited or even no WYLD."

9. L500 – equation 7

 **Authors' Response:thank you very much for pointing out this error**, and it will be corrected in the revised manuscript.

10. L560 – "streamflow photograph"? hydrograph?

 **Authors' Response:thank you very much for pointing out this typo**, and it will be corrected in the revised manuscript.

[Figure]

**Fig. 1.** Spatial variation of annual precipitation at sub-basin scale for (a) Gauge (b) CHIRPS and (c) CPC.

**Fig. 2.** Observed and simulated discharges at the outlet of JRW at monthly scale using precipitation inputs of Gauge, CHIRPS and CPC, respectively.

[Figure]

**Fig. 3.** Observed and simulated discharges at the outlet of JRW at daily scale using precipitation inputs of Gauge, CHIRPS and CPC, respectively.

[Figure]

**Fig. 4.** Water balance components for all sub-basins derived from SWAT models using precipitation inputs of (a) Gauge (b) CHIRPS and (c) CPC at monthly scale and (d) Gauge (e) CHIRPS and (f) CPC at daily scale

---

## Author Response (AR1)

**Author's Response**

**Manuscript Number:** hess-2020-56

**Authors:** Jianzhuang Pang, Huilan Zhang, Quanxi Xu, Yujie Wang, Yunqi Wang, Ouyang Zhang, Jiaxin Hao

**Title:** Hydrological evaluation of open-access precipitation data using SWAT at multiple temporal and spatial scales

we sincerely appreciate the comments on our manuscript. We both thank the positive remarks and the specific concerns, which provided great encouragement and specific guidance to the authors to improve the manuscript. Please see the bellowing point-to-point responses to the main concerns and minor comments.
* * *
**Response to the reviews**

**Response to Comments from Reviewer #1**

**General Comments:** *This paper uses two open-access precipitation products (CHIRPS and CPC) and a dataset from rain gauges to drive the SWAT hydrological model in the Jiang river watershed in China. All three precipitation datasets are shown to produce generally similar hydrological model performances, with the calibrated parameterisation reducing the effect of the identified differences in the precipitation datasets through changing hydrological processes. This is a potentially useful paper for the hydrological modelling community in that it highlights that an acceptable hydrological performance according to the commonly-used Moriasi et al (2007) criteria for the Nash Sutcliffe Efficiency metric does not mean that the hydrological processes are correctly simulated – but that it can indicate that the calibration process has merely been successful in altering the catchment hydrological processes to compensate for inadequacies in the input data.*

**Main concerns:**

*1. The paper fails to articulate the implications of its finding (that hydrological models can give very similar model performance, with differing process behaviour, with precipitation datasets with quite different characteristics) in either the Conclusions or the Abstract. For example, Remesan and Holman (2015) study cited by the authors showed that such 'similar' calibrated/validated models, when subsequently run using perturbed inputs (e.g. climate change scenario), can lead to different magnitudes and directions of hydrological change due to their differing parameterization. The authors should consider how their findings can guide modelers in the use of these different precipitation datasets for the hydrological modelling of the current and future climate.*

**Authors' response**: Greatly appreciate the comment and suggestion. As stated in Remesan and Holman's (2015) study, "with similar historical model performance, model construction with different baseline meteorological data choices significantly condition the magnitude and direction of simulated hydrological impacts of climate change", the current study has reached "similar" conclusions: "with similar performances in simulating river runoff, different types of precipitation data digested in hydrologic modeling tends to counterbalance their identified differences by differing parameterization and leads to different directions of hydrologic processes". Considering that this research focuses on precipitation condition under current climate, it could generally provide implications to hydrological modelers of current and future climate from following two aspects:

1) From perspective of precipitation estimation: CHIRPS has a higher spatial resolution (with 0.05° being equivalent to a resolution of one gauge station for every 30.25 $km^2$ area) and a stronger ability to recognize heavy rain and extreme rainfall (Fig.4 – Fig.6 in the manuscript). These features would facilitate the widespread use of CHIRPS in future climate analyses. Take extreme climate analyses for example, it is reported that the frequency of extreme rainfall events in China has been significantly increased in past decades and this tendency will continue increasing in future climate change (Mou et al., 2020; Xi et al., 2018). With this background in future climate change, CHIRPS would provide high potential in future extreme rainfall event analyses with high spatial resolution. Actually, CHIRPS has been applied to identify extreme rainfall events by indicators of nP (Number of

days with $P \geqslant 1mm$), PRCPTOT (Annual total precipitation), and R95pad (Total precipitation when $P > 95$ percentile of all days), etc. (Cavalcante et al., 2020). In contrast, the CPC's strong ability to identify light rain represents a unique advantage in extreme drought-related research.

2) From perspective of hydrologic modeling: overall, the three precipitation types derive almost equivalent and acceptable hydrological performance according to Moriasi et al's criteria (2007), while CHIRPS presented better performance in uncertainty analyses. Although the river runoff values simulated by the three models are basically consistent, there are significant differences among other hydrological components, such as surface runoff, lateral flow, and base flow. CHIRPS tends to derive more surface flow due to the higher precipitation detection, while CPC tends to yield more lateral flow due to the lower precipitation detection. As such, CHIRPS would suit broader applications in flood prediction of the future climate due to its ability in extreme precipitation identification and surface flow simulation. More importantly, multiple-objective calibration based on multiple hydrological components are recommended to improve SWAT modeling in large and spatial resolved watershed.

Cavalcante, R. B. L., Ferreira, D. B. da S., Pontes, P. R. M., Tedeschi, R. G., da Costa, C. P. W., & de Souza, E. B. (2020). Evaluation of extreme rainfall indices from CHIRPS precipitation estimates over the Brazilian Amazonia. Atmos. Res., 238, 104879. doi:10.1016/j.atmosres.2020.104879.
Mou, S., Shi, P., Qu, S., Feng, Y., Chen, C., & Dong, F. (2020). Projected regional responses of precipitation extremes and their joint probabilistic behaviors to climate change in the upper and middle reaches of Huaihe River Basin, China. Atmos. Res., 104942. doi:10.1016/j.atmosres.2020.104942
Xi, Y., Miao, C., Wu, J., Duan, Q., Lei, X., & Li, H. (2018). Spatiotemporal Changes in Extreme Temperature and Precipitation Events in the Three-Rivers Headwater Region, China. J. Geophys. Res-Atmos., 123, 5827–5844. doi:10.1029/2017jd028226.

The above considerations were articulated in sections of **Abstract**, **Discussion** and **Conclusions of the revised manuscript**. Supplements relating to the first aspect are marked blue, while those relating to the second aspect are marked red.

[revised manuscript text omitted]

*2. Given that the authors are simulating a 159,000km² catchment using a single flow gauge for calibration / validation, there is huge equifinality in their results. Given that they used the SUFI-2 / SWATCUP, I would have expected some assessment and discussion of the uncertainty in their model results.*

**Authors' response**: Greatly appreciate the comment. Assessment and discussions on the uncertainty of model results are quite important issues in hydrologic modelling (Abbaspour, 2015). In our study, the model calibration / validation use a single hydrologic station, with a monitored area of more than 159000 km², which would induced inevitable system or random deviation by parameter calibration. Therefore, as the comment suggested, uncertainty analyses on model results should be processed and discussed.

Abbaspour, K. C. (2015) SWAT-CUP 2012: SWAT Calibration and Uncertainty Programs - A User Manual. Tech. rep., Swiss Federal In-stitute of Aquatic Science and Technology, Eawag, Dübendorf, Switzerland.

With the considerations above, assessment and discussion on the uncertainty of model results was added in the revised manuscript, and the modification was specified as follows:

[revised manuscript text omitted]

*3. The paper provides three sets of SWAT output analyses – monthly, daily and daily aggregated to monthly. However, SWAT is a daily model so the monthly SWAT outputs are themselves an internal aggregation of its daily outputs; so the presentation and description of the daily aggregated to monthly outputs (L439-448 and Figures 12 and 13) are meaningless and should be removed.*

    **Authors' Response:** Thanks a lot for this comment and advice. The presentation and description of the daily aggregated to monthly outputs (L439-448 and Figures 12 and 13) was removed in the revised manuscript.

    As one of the major objectives of this manuscript was to evaluate the performances of different precipitation datasets in simulating the watershed streamflow using SWAT on different temporal scales, the authors ran the SWAT models at monthly and daily scales, respectively. Essentially, SWAT is a daily model that monthly outputs can be derived by aggregating its daily outputs. For researchers, who are not able to collect daily streamflow records, may be more interested in the performance at monthly scale. With this consideration, the authors presented two sets of SWAT output analyses, i.e. daily and monthly, and further look into the corresponding water balance components (Fig.4 & Table 5) adjusting by calibrated parameters (Table 2). In the previous manuscript, proportions of water balance components at monthly scale were compared and analyzed. In the revised manuscript, water balance components calculated at daily scale should also be presented and compared with results of monthly-scaled models.

**Figure 16.** Water balance components for all sub-basins derived from SWAT models using precipitation inputs of (a) Gauge (b) CHIRPS and (c) CPC at monthly scale and (d) Gauge (e) CHIRPS and (f) CPC at daily scale (where SURQ represents surface runoff $Q_{surf}$; LATQ represents lateral flow $Q_{lat}$; GW_Q is the baseflow from the shallow aquifer; GW_Q_D is the baseflow from the deep aquifer, and the sum of GW_Q and GW_Q_D equals to $Q_{gw}$; ET represents actual evapotranspiration $ET$.

**Table 2: Optimal parameters calibrated for all three models.** (excerpts)

| Parameters | Initial range | Gauge Monthly | Daily | CHIRPS Monthly | Daily | CPC Monthly | Daily |
|---|---|---|---|---|---|---|---|
| a__SOL_K().sol | −10/10 | 1.988/10 | -0.706/10 | -0.471/7.681 | -0.396/10 | 5.264/10 | -2.106/10 |
| v__ESCO.hru | 0/1 | 0.879/1 | 0.405/1 | 0.775/1 | 0.355/1 | 0.914/1 | 0.462/1 |
| v__ALPHA_BF.gw | 0/1 | 0.401/0.963 | 0.299/0.896 | 0.055/0.677 | 0.183/0.728 | 0.216/0.901 | 0.415/1 |

**Table 5: Summarization of annual average water balance components of the three models for the whole JRW.**

| Time scale | Datasets | Statistics | SURQ | LATQ | GW_Q | GW_Q_D | ET | Summation |
|---|---|---|---|---|---|---|---|---|
| Monthly | Gauge | Average amount/mm | 4500.00 | 2977.22 | 299.07 | 60.61 | 9076.60 | 16913.50 |
| | | Percentage/% | 26.61% | 17.60% | 1.77% | 0.36% | 53.66% | |
| | CHIRPS | Average amount/mm | 6068.35 | 773.24 | 949.56 | 140.79 | 9046.83 | 16978.78 |
| | | Percentage/% | 35.74% | 4.55% | 5.59% | 0.83% | 53.28% | |
| | CPC | Average amount/mm | 1087.19 | 5577.20 | 583.45 | 30.15 | 8694.40 | 15972.40 |
| | | Percentage/% | 6.81% | 34.92% | 3.65% | 0.19% | 54.43% | |
| Daily | Gauge | Average amount/mm | 5544.88 | 1856.00 | 244.94 | 48.29 | 9309.37 | 17003.48 |
| | | Percentage/% | 32.61% | 10.92% | 1.44% | 0.28% | 54.75% | |
| | CHIRPS | Average amount/mm | 6202.63 | 834.78 | 1167.37 | 59.75 | 10434.58 | 18699.11 |
| | | Percentage/% | 33.17% | 4.46% | 6.24% | 0.32% | 55.80% | |
| | CPC | Average amount/mm | 2493.11 | 2302.28 | 1709.95 | 88.66 | 9384.90 | 15978.90 |
| | | Percentage/% | 15.60% | 14.41% | 10.70% | 0.55% | 58.73% | |

**Results showed that:**

    (1) Either at daily scale or monthly scale, all three models achieved acceptable and similar simulation performance for comparisons of both time series and spatial distributions. However, the parameter systems are completely different at two temporal scales (Table 2). The non-uniqueness of parameters has been proved a persistent drawback of SWAT (Abbaspour et al., 2004; Abbaspour, 2015; Zhang et al., 2015).

    And we had explained this drawback at **line399** to **line404** in manuscript:

    "*In general, simulated and observed streamflow hydrographs, using OPPs and Gauge inputs, can successfully match at both monthly and daily scales. However, consistency between simulated and observed streamflow does not guarantee identical hydrologic processes. For example, the SWAT model calibrated parameters are not the same for all precipitation inputs, meaning that the hydrologic mechanics during SWAT modelling are also different. As such, it is critical that researchers and decision makers adequately understand the benefits and limitations of different precipitation products in modelling the hydrologic processes.*"

    (2) With differing parameterizations, different precipitation inputs tend to derive completely different hydrological component amounts at different time scales (Fig. 16 & Table 5).

Abbaspour, K. C., Johnson, C. A., & van Genuchten, M. T. (2004). Estimating Uncertain Flow and Transport Parameters Using a Sequential Uncertainty Fitting Procedure. Vadose Zone Journal, 3(4), 1340. doi:10.2136/vzj2004.1340

Abbaspour, K. C. (2015) SWAT-CUP 2012: SWAT Calibration and Uncertainty Programs - A User Manual. Tech. rep., Swiss Federal In-stitute of Aquatic Science and Technology, Eawag, Dübendorf, Switzerland.

Zhang, J., Li, Q., Guo, B., & Gong, H. (2015). The comparative study of multi-site uncertainty evaluation method based on SWAT model. Hydrological Processes, 29(13), 2994–3009. doi:10.1002/hyp.10380.

With the considerations above, discussion on the model parameters and water balance components was added in the revised manuscript, and the modification was specified as follows:

**Original version of discussion section:**

[revised manuscript text omitted]

**Minor revision comments:**

1. L19 – change "All three products" to "Both OPPs" as the text is comparing to the gauge model.

    **Authors' Response: Thanks a lot for pointing out this issue.**

    P1-L17 – "All three products satisfactorily reproduce the stream discharges at the JRW outlet with better performance than the Gauge model." was corrected as "Both OPPs satisfactorily reproduce the stream discharges at the JRW outlet with slightly worse performance than the Gauge model, …"

2. L153 – is the evapotranspiration "actual", "potential" or "reference"?

    **Authors' Response: It's the actual evapotranspiration.**

    P5-L123 – "the annual average evapotranspiration ranges from 800 to 1000 mm." was revised as "the annual average actual evapotranspiration (ET) ranges from 800 to 1000 mm."

    The descriptions related to evapotranspiration all through the manuscript have been corrected in the revised manuscript:

    P10-L214 – "Water balance, including precipitation, surface runoff, evapotranspiration, infiltration, lateral and base flow, and percolation to shallow and deep aquifers, is mathematically expressed as follows:"

    The sentence was corrected as "Water balance, including precipitation, surface runoff, actual evapotranspiration, infiltration, lateral and base flow, and percolation to shallow and deep aquifers, is mathematically expressed as follows:"

    P10-L217 – "$E_a$ = evapotranspiration" was corrected as "$ET$ = actual evapotranspiration".

3. L169-170 – how has the classification accuracy been determined, given that it was based on "manual visual interpretation"?

    **Authors' Response: Thank you very much for this question.**

    The procedure of deriving LUCC types based on 2010 Landsat TM/ETM remote sensing images are as follows: The geometric shape, colour feature, texture feature and spatial distribution of ground objects were analysed and extracted according to the image spectral features. The remote sensing image interpretation marks were established based on the field measurement data and the reference map. Six primary classifications were recognized- cultivated land, woodland, grassland, water area, construction land, and unused land. The quality of the LUCC product was checked by combining field survey and random sampling dynamic map spot for repeated interpretation analysis. Generally, the quality inspection result is that the classification accuracy of cultivated land data is ~85%, and that of other data can reach more than 75%.

    P6-L135-137 – "The data included six primary classifications—cultivated land, woodland, grassland, water area, construction land, and unused land, as well as 25 secondary classifications. The cultivated land's classification accuracy was 85 %, and other data classification accuracies reached 75 %."

    The sentence was revised as "The data included six primary classifications—cultivated land, woodland, grassland, water area, construction land, and unused land, as well as 25 secondary classifications. After checking the quality of data products by combining field survey and random sampling dynamic map spot for repeated interpretation analysis, it is proved that the cultivated land's classification accuracy was 85 %, and other data classification accuracies reached 75 %.".

4. L194 – how does a dataset (CHIRPS v2.0) released in 2015 provide data to the "present"?

    **Authors' Response: Thank you very much for this question.**

    Actually, the CHIRPS v2.0 dataset has been continuously updated since it was released in 2015, and we are sorry for the misinterpretation.

    P7-L156 – "The most recent gridded format CHIRPS product (V2.0 datasets) was completed and released in February 2015."

    The sentence was revised as "The first gridded format CHIRPS product was released in February 2015, which has first recorded in 1981 and continues to be updated."

5. L237 – looking at equation (3), isn't the optimal value of $STD_n$ = 1 e.g. identical $STD$s? And why should $STD_n$ values range from 0-1 which implies $STD$ gauge can never be < $STD$ opp? General –$RMSE$, $STD$ and $PBIAS$ have units – please use them throughout

    **Authors' Response: we are sorry to make this mistake for our neglect.**

    P9-L193 – "The $STD_n$ values range from 0 to 1, and the optimal value is 0. The $STD$ value is mathematically expressed as follows:"

The sentence should be corrected as: "The $STD_n$ values range from 0 to ∞, and the optimal value is 1. The $STD_n$ value is mathematically expressed as follows:"

The units of *STD* and *PBIAS* was revised throughout the manuscript as follows:

P8-L189-190 – "Where *n* is the number of the time series; $Q_i$ and $S_i$ are measured values and estimated values (or simulated values), respectively; and $\overline{Q}$ and $\overline{s}$ are the mean values of the measured and estimated values (or simulated values), respectively."

The sentence was revised as "Where *n* is the number of the time series; $Q_i$ and $S_i$ are measured values and estimated values, respectively; and $\overline{Q}$ and $\overline{s}$ are the mean values of the measured and estimated values, respectively. The value may refer to either precipitation (mm) or streamflow discharge (m$^3$/s)."

P9-L191 – "Standard deviation (*STD*) represents the discretization degree of the datasets."
The sentence was revised as "Standard deviation (*STD*) represents the discretization degree of the precipitation datasets (mm)."

P12-L248 – "*PBIAS* describes the OPPs' systematic bias. *PBIAS* ranges from 0 to +∞, and the optimal value is 0."
The sentence was revised as "*PBIAS* describes the OPPs' systematic bias (%). *PBIAS* ranges from 0 to +∞ %, and the optimal value is 0 %."

P12-L266 – "The *STD* values for Gauge-CHIRPS and Gauge-CPC are 1.06 and 0.94, respectively."
The sentence was revised as "The $STD_n$ values for Gauge-CHIRPS and Gauge-CPC are 1.06 and 0.94, respectively."

P13-L268-270 – "Nevertheless, *PBIAS* values of Gauge-CHIRPS and Gauge-CPC were 9.58 and -6.70, respectively"
The sentence was revised as "Nevertheless, *PBIAS* values of Gauge-CHIRPS and Gauge-CPC were 9.58 % and -6.70 %, respectively"

P16-L343-344 – "The *PBIAS* values for Gauge-CHIRPS and Gauge-CPC are 5.85 and -5.38, respectively."
The sentence was revised as "The *PBIAS* values for Gauge-CHIRPS and Gauge-CPC are 5.85 % and -5.38 %, respectively."

6. L463 – "antecedent" is the more usual term for "early-stage".
   **Authors' Response: Thank you very much for your advice, and we have revised this term into antecedent all through the manuscript:**
   P17-L373-375 – "Solano-Rivera et al. (2019) experimented in the San Lorencito headwater catchment and found that the rainfall-runoff dynamics before extreme events were mainly related to early-stage conditions. After extreme flood events, early-stage conditions had no effect on rainfall-runoff processes, and rainfall significantly affected the streamflow discharge."
   The sentence was changed as "Solano-Rivera et al. (2019) experimented in the San Lorencito headwater catchment and found that the rainfall-runoff dynamics before extreme events were mainly related to antecedent conditions. After extreme flood events, antecedent conditions had no effect on rainfall-runoff processes, and rainfall significantly affected the streamflow discharge."

7. L483 – there are no ALPHA-BF parameter ranges given in Table 1 and 2 to substantiate this. The values of ALHPA_BF and GWRECH_DP should be added to the tables.
   **Authors' Response:** Thanks a lot for pointing out this error, and ALHPA_BF has been added to Table1 and Table2. Since the parameter RCHRG_DP is not a sensitive one, so it was not included in the calibration process.

**Table 1: Hydrological parameters considered for sensitivity analysis ("a_", "v_" and r_" means an absolute increase, a replacement, and a relative change to the initial parameter values, respectively). (excerpts)**

| Parameters | Description | Range | Default |
|---|---|---|---|
| v__PLAPS.sub | Precipitation lapse rate[mm] | -1000/1000 | 0 |
| v__ALPHA_BF.gw | Baseflow alpha factor [days$^{-1}$] | 0/1 | 0.048 |
| v__ALPHA_BNK.rte | Baseflow alpha factor for bank storage | 0/1 | 0 |

**Table 2: Optimal parameters calibrated for all three models.** (excerpts)

| Parameters | Initial range | Gauge Monthly | Daily | CHIRPS Monthly | Daily | CPC Monthly | Daily |
|---|---|---|---|---|---|---|---|
| v__PLAPS.sub | −1000/1000 | 0.012/0.067 | 0.061/0.183 | 0.079/0.135 | 0.068/0.205 | 0.017/0.078 | -0.014/0.095 |
| v__ALPHA_BF.gw | 0/1 | 0.401/0.963 | 0.299/0.896 | 0.055/0.677 | 0.183/0.728 | 0.216/0.901 | 0.415/1 |
| v__ALPHA_BNK.rte | 0/1 | 0.492/0.863 | 0.444/1 | 0.201/0.696 | 0.467/1 | 0.564/1 | 0.307/0.92 |

8. L486 – what is "proletarian" flow?

**Authors' Response:**

Thank you so much for pointing out this typo. The authors tended to articulate that "For CPC dataset, the high proportion of LR events will lead to severe rainfall losses in the initial- and post- loss processes, resulting in very limited surface water yield.

As such, "A potential reason for this phenomenon may be that the rainfall during LR events tends to be easily lost in the initial- and post- loss processes, resulting in low proletarian flow and thus WYLD." was corrected as:

P18-L392-393 – "A potential reason for this phenomenon may be that the rainfall during LR events tends to be easily lost in the initial- and post-loss processes, resulting in very limited or even no WYLD."

9. L500 – equation 7

**Authors' Response:  thank you very much for pointing out this error.**

P18-L405 – "According to the SWAT model's water balance equation (Eq. 9), …" was corrected as "According to the SWAT model's water balance equation (Eq. 7), …"

10. L560 – "streamflow photograph"? hydrograph?

**Authors' Response:  thank you very much for pointing out this typo.**

P20-L452 – "The differences in the statistics at the monthly and daily scale correspondingly affected the streamflow photograph, …" was corrected as "However, the differences of precipitation inputs in the statistics at the monthly and daily scale correspondingly affected the streamflow hydrograph, …"

**Response to Comments from Reviewer #2**

**General comment:** *This paper presents a hydrological evaluation of two open-access precipitation products (CHIRPS and CPC) compared with rain gauge dataset, at multiple temporal and spatial scales. The content of this research is of great interest to readers of watershed hydrology, remote sensing, and satellite meteorology, since it provided valuable suggestions for researchers in these fields, especially for hydrologic modelers. It is demonstrated by the authors that, even with obvious statistical differences, performances of the three selected precipitation datasets in simulating water yield are parallel. Comparably, inconsistency were found when OPPs and rain gauge data were used to simulate hydrological components, e.g. Surface runoff, lateral flow, and base flow. Inner mechanism was highlighted from both spatial and temporal scales. Overall, this manuscript is quite well written and presented. Minor revision comments below aim to improve the quality of the manuscript.*

**Main concerns:**

*1. The spatial resolutions of CHIRPS (0.05°) and CPC (0.5°) were higher than that of the geographic datasets, "some of the grid records are potentially missed, especially for the high - resolution CHIRPS products." Duan Z et al. (2019) proposed an area-weighted method to calculated precipitation for each subbasin, "Calculate the area-weighted average daily CHIRPS data (after disaggregated by 10 times (0.005°)) from all grids within the subbasin to represent the effective daily precipitation for each subbasin". This might be an alternative way to solve the data problem.*

**Authors' response**: Greatly appreciate your suggestion. Following this advice, we recalculated the precipitation inputs for each sub-basin via "area-weighted" (AW) method, and compared the results with those derived by "Nearest Distance" (ND) adopted in this manuscript. Take the sub-basin No.411, which is located at Beibei hydrological station, as an example, the calculated precipitation inputs by the above two methods were depicted in Fig. S1. It's shown in Fig.S1 that no significant difference ($P = 0.88$) were detected by using these two methods. Compared with the ND method, the amount of rainfall obtained by AW method is slightly underestimated ($PBIAS$ = -0.78%), especially when the rainfall intensity is between 50 mm and 100mm. Both methods slightly increase the uncertainty of precipitation inputs. Considering the effectiveness in producing peak discharges of streamflow in SWAT model, the ND method in the original manuscript is adopted. Actually, the "potentially missed" grid records mainly refers to CHIRPS products. More than 10000 grids within the JRW or nearby were adopted, which is much higher than that of the CPC (~165 grids) and rain gauges (20 stations). Considering that the "missed" grids could be covered by other grids within the same sub-basin due to the high resolution of CHIRPS.

[Figure]

**Figure S1.** Scatter plot of the CHIRPS precipitation comparing between Nearest Distance method and Area Weighted method.

Therefore, to avoid misrepresentation of data accuracy, the manuscript will be revised as follows (**line 177** to **line 178**):

**Original version of abstract:**
P8-L177-178 – "Using this method, some of the grid records are potentially missed, especially for the high-resolution CHIRPS products."

**Revised version of abstract:**

Using this method, the grid records of high-resolution CHIRPS products within the same sub-basin will be uniformly assigned the grid value closest to the centroid, which will offset the high resolution advantage of CHIRPS products.

*2. Moriasi et al. (2007), cited by the author, used three indicators RSR (ratio of the root mean square error to the standard deviation of measured data), NSE, and PBIAS to establish a model to evaluate performance level, while the author used only two. Why not use all three metrics? Besides, since only the NSE index was graded into different evaluation levels, was the evaluation on model performance reliable without the evaluation grades from other two indicators?*

**Authors' response**: Greatly appreciate the comment. After an extensive literature review on model evaluation system, we followed the suggestion and adopted all three indicators to evaluate the performance of the hydrologic models with different precipitation datasets. The models' performances were classified using the standard defined by Moriasi et al. (2007), which are shown in Table 2. And the evaluation results by using all three metrics at monthly and daily scales were depicted in Table S1 and Table S2, respectively.

Comparably, in the original manuscript, like most existing papers did (Zhu et al., 2015; Tuo et al., 2018; Duan et al., 2019), only *NSE*, *CC*, and *PBIAS* were adopted to evaluate the simulation performance of hydrologic models. But they did not give a classified evaluation of the simulation results. When we used only *NSE* to classify the simulation performance of the model, as presented in the original manuscript, the evaluation results are "Very good" for all three models at the monthly scale, but when we used three indicators to evaluate, CPC only reached the level of "Good" (Table S1.).

Table 2. General performance ratings statistics recommended by Moriasi et al. (2007).

| Performance Rating | *RSR* | *NSE* | *PBIAS* (%) |
|---|---|---|---|
| Very good | 0.00 < RSR ≤ 0.50 | 0.75 < NSE ≤ 1.00 | PBIAS ≤ ±10 |
| Good | 0.50 < RSR ≤ 0.60 | 0.65 < NSE ≤ 0.75 | ±10 < PBIAS ≤ ±15 |
| Satisfactory | 0.60 < RSR ≤ 0.70 | 0.50 < NSE ≤ 0.65 | ±15 < PBIAS ≤ ±25 |
| Unsatisfactory | RSR > 0.70 | NSE < 0.50 | PBIAS > ±25 |

Table S1. Evaluation results of monthly scale SWAT model

| | Calibration | | | Validation | | |
|---|---|---|---|---|---|---|
| | Gauge | CHIRPS | CPC | Gauge | CHIRPS | CPC |
| *RSR* | 0.28 | 0.42 | 0.36 | 0.36 | 0.49 | 0.46 |
| *NSE* | 0.92 | 0.82 | 0.87 | 0.87 | 0.76 | 0.79 |
| *PBIAS* (%) | 7.9 | 2.3 | 10.8 | 1.2 | -6.6 | 4.2 |
| Evaluation | Very good | Very good | Good | Very good | Very good | Very good |

Table S2. Evaluation results of daily scale SWAT model

| | Calibration | | | Validation | | |
|---|---|---|---|---|---|---|
| | Gauge | CHIRPS | CPC | Gauge | CHIRPS | CPC |
| *RSR* | 0.52 | 0.67 | 0.67 | 0.53 | 0.69 | 0.63 |
| *NSE* | 0.73 | 0.55 | 0.55 | 0.72 | 0.53 | 0.6 |
| *PBIAS* (%) | 5.1 | -1.7 | 19.9 | 0.2 | -12.2 | 9.9 |
| Evaluation | Good | Satisfactory | Satisfactory | Good | Satisfactory | Satisfactory |

Duan, Z., Tuo, Y., Liu, J., Gao, H., Song, X., Zhang, Z., Yang, L., and Mekonnen, D. F.: Hydrological evaluation of open-access precipitation and air temperature datasets using SWAT in a poorly gauged basin in Ethiopia, J. Hydrol., 569, 612-626, https://doi.org/10.1016/j.jhydrol.2018.12.026, 2019.

Tuo, Y., Marcolini, G., Disse, M., and Chiogna, G.: A multi-objective approach to improve SWAT model calibration in alpine catchments, J. Hydrol., 559, 347-360, https://doi.org/10.1016/j.jhydrol.2018.02.055, 2018.

Zhu, H., Li, Y., Liu, Z., Shi, X., Fu, B., and Xing, Z.: Using SWAT to simulate streamflow in Huifa River basin with ground and Fengyun precipitation data, J. Hydroinform., 17, 834–844, https://doi.org/10.2166/hydro.2015.104, 2015.

The above considerations was supplemented in sections of **Materials and methods** and **Result** of the revised manuscript.

**Original version of materials and methods section 2.3:**
P9-L195-196 – "(3) *RMSD* value: Root mean square deviation (*RMSD*) is used to demonstrate the error

between the OPPs and Gauge datasets (mm). *RMSD* has a range from 0 to +∞ mm, and an optimal value of 0

mm. The *RMSD* value is expressed as follows:"

$$RMSD = \sqrt{\frac{1}{n}\sum_{i=1}^{n}\left[(Si-\overline{S})-(Qi-\overline{Q})\right]^2} \text{ ,} \qquad (4)$$

**Revised version of materials and methods section 2.3:**
(3) *RSR* value: observations standard deviation ratio (*RSR*) is an error index statistic between the OPPs and Gauge datasets. Root mean square error (*RMSE*) divided by *STD* values would derive the *RSR* value. *RSR has a* range from 0 to ∞ with 0 as the optimal value. The calculation equation is expressed as follows:

$$RSR = \frac{RMSE}{STD} = \frac{\sqrt{\sum_{i=1}^{n}(S_i-Q_i)^2}}{\sqrt{\sum_{i=1}^{n}(S_i-\overline{S})^2}}, \qquad (4)$$

**Original version of materials and methods section 2.4.2:**
P11-L244-247 – "The model performance was classified using the *NSE* values defined by Moriasi et al. (2007): unsatisfactory performance (*NSE* ≤ 0.50), satisfactory performance (0.50 < *NSE* ≤ 0.65), good performance (0.65 < *NSE* ≤ 0.75) and very good performance (0.75 < *NSE* ≤ 1.00). The three models' parameter range after 3000 iterations is shown in Table 2. CC, *NSE*, and Percentage bias (*PBIAS*) were used to evaluate the model simulation results."

**Revised version of materials and methods section 2.4.2:**
The model performance was classified using *RSR*, *NSE*, and percentage bias (*PBIAS*) values defined by Moriasi et al. (2007), which is shown in Table 2. The three models' parameter range after 3000 iterations is shown in Table 3. CC, *NSE*, *PBIAS and RSR* were used to evaluate the model simulation results.

**Original version of result section:**
P15-L325-328 – "Based on the model performance classification scheme designed by Moriasi et al. (2007), all three models, each using a different precipitation product, achieved "very good" performance for both the calibration and verification periods, although the Gauge model attained the highest *CC* (0.93 for calibration and 0.87 for validation) and *NSE* (0.92 and 0.87)."

**Revised version of result section:**
Based on the model performance classification scheme designed by Moriasi et al. (2007), Gauge and CHIRPS achieved "very good" performance for both the calibration and verification periods, although the Gauge model attained the highest *NSE* (0.92 for calibration and 0.87 for validation) values and lowest *RSR* (0.28 and 0.36) value, while CPC only reached the level of "Good" due to higher *PBIAS* (10.8 %) (Fig.9).

*3. Explain what "The correlation coefficients' spatial variation" is? The spatial correlation of the three precipitation datasets should be a value rather than a graph. Explain how Fig. 8 was calculated and obtained. Explain why distinguish average precipitation in daily and monthly scales?*

**Authors' response**: Thanks a lot for the question. Basically, the term "correlation coefficients" ($CC_{sub}$) refers to the correlation between two OPPs and Gauge at either daily or monthly scales for each sub-basin. "The correlation coefficients' spatial variation" is the variation of $CC_{cub}$ in different sub-basins. The calculation steps of $CC_{sub}$ are as follows. Firstly, the spatial distribution diagrams of Gauge records and the two OPPs are calculated. The correlation coefficient for each sub-basin between the Gauge series and OPPs series is calculated by using the correlation coefficient method. Coefficients for the all 414 sub-basins will form a spatial distribution plot of the correlation coefficients. Fig. 8 aims to distinguish the correlation of rainfall amounts in different sub-basins, and to preliminarily judge the performance of OPPs in different sub-basins, especially at different time scales. Distinguish average precipitation at daily and monthly scales are important, since the difference in precipitation amount statistics may highly resulted in different modelling performance and inner mechanics of hydrologic processes. Actually, the $CC$ values between spatial-aggregated CPC and Gauge records are higher than that of CHIRPS and Gauge at both scales, yet the $CC$ values at the monthly scale are much higher than that at daily scale. Similar variation regularity was found in spatial distribution, which was described in Fig. 8. However, $CC$ values between CPC and Gauge are smaller than that of CHIRPS and Gauge at a portion of sub-basins, which was explained at **line313** to **line315** in manuscript:

 *"Spatially, the higher CC values between the Gauge and CPC at the monthly scale are mainly distributed in areas with comparably low or high rainfall amounts, such as Wudu and Wangyuan. Yet, the CC value was less relevant in areas with moderate rainfall (e.g., Suining), compared with to that of the Gauge and CHIRPS."*

The manuscript will be accordingly revised as follows (**line 311** to **line 315**). Additions to the spatial correlation $CC$ are marked blue, and additions to the correlation coefficients' spatial variation are marked red:

**Original version:**
 P14-L311-312 – "The correlation coefficients' spatial variation between the Gauge and OPPs at monthly and daily scales are illustrated in Fig. 8."

**Revised version:**
 "The $CC$, $STDn$, and $RSR$ values of precipitation spatial distribution between CHIRPS and Gauge are 0.89, 0.96, and 0.55, respectively, and 0.82, 0.87, 0.62 between CPC and Gauge, respectively. These statistics indicate that both CHIRPS and CPC estimates can describe the spatial distribution of precipitation in JRW, among which CHIRPS depicts better performance. The correlation coefficients between the Gauge and OPPs at monthly or daily scales for every sub-basin are illustrated in Fig. 8.

**Minor revision comments:**

1. Double-check: L11- "Jiang River". L114- "larges" should be "largest". L299- "varations" should be "variations".

    **Authors' Response: Thanks a lot for pointing out the typos, and we made revisions all through the manuscripts.**

    P1-L11-12 – "Herein, we implemented a comprehensive evaluation of three selected precipitation products in the Jiang River Watershed (JRW) located in southwest China." was corrected as "Herein, we implemented a comprehensive evaluation of three selected precipitation products in the Jialing River Watershed (JRW) located in southwest China."

    P5-L114 – "The Jialing River is the primary tributary of the Yangtze River, with the larges drainage area of 159,812 km$^2$ and a total length of ~1345 kilometers." was corrected as "The Jialing River is the primary tributary of the Yangtze River, with the largest drainage area of 159,812 km$^2$ and a total length of ~1345 kilometers."

    P14-L299 – "Spatial varations of the three products' long-term mean annual precipitation, …" was corrected as "Spatial variations of the three products' long-term mean annual precipitation, …"

2. Grammar errors. L98- The verb "have" should be "has". L92- The article "an" here should be "a". L310- "relative" should be "relatively".

    **Authors' Response: Thank you very much for pointing out the grammar errors, and we have corrected them as follows:**

    P4-L98 – "However, evidence for water balance component variations under the influence of different precipitation inputs have not been fully investigated." was corrected as "However, evidence for water balance component variations under the influence of different precipitation inputs has not been fully investigated."

    P4-L92 – "Bai & Liu (2018) used an HIMS model …" was corrected as "Bai & Liu (2018) used a HIMS model…"

    P14-L310 – "the overall precipitation values estimated by the CHIRPS are relative higher" was corrected as "the overall precipitation values estimated by the CHIRPS are relatively higher"

3. L342 & L360-361 unit of CC should be decimal rather than percentage.

    **Authors' Response: Thank you so much for the suggestion.**

    P16-L342 – "The spatial correlation between WYLD and precipitation for rainfall for the Gauge, CHIRPS, and CPC products reached 84.8 %, 84.3 %, and 90.84 %, respectively."
    The sentence was corrected as "The spatial correlation between WYLD and precipitation for rainfall for the Gauge, CHIRPS, and CPC products reached 0.85, 0.84, and 0.91, respectively."

    P16-L360-361 – "The *CC* values between the WYLD and precipitation for Gauge, CHIRPS, and CPC at the daily scale are 83.45 %, 84.41 % and 91.70 %, respectively, …"
    The sentence was corrected as "The *CC* values between the WYLD and precipitation for Gauge, CHIRPS, and CPC at the daily scale are 0.83, 0.84 and 0.92, respectively, …"

4. L257-As IPCC reported, "Extreme rainfall" was defined as the 95th percentile of daily precipitation data. Therefore, Fig.3, shown as monthly rainfall box chart, failed to capture "extreme rainfall values"

    **Authors' Response: Thanks a lot for the comment and suggestion.**

    The term "extreme rainfall" here means that estimation of Gauge monthly rainfall in the rainy season (especially in July) is significantly higher than that of the other two OPPs. For better interpretation, the term will be revised as:

    P12-L257 – "Fig. 3 shows that the extreme rainfall values captured by Gauge are higher than those of CHIRPS and CPC." was corrected as "Fig. 3 shows that the rainfall values in the rainy season (especially in July) captured by Gauge are higher than those of CHIRPS and CPC."

5. L327-Usually we use "validation" instead of "verification".

    **Authors' Response: Thank you very much for your advice, and we have revised this term into validation all through the manuscript:**

P15-L327 – "Based on the model performance classification scheme designed by Moriasi et al. (2007), all three models, each using a different precipitation product, achieved "very good" performance for both the calibration and verification periods, …"

The sentence was corrected as "Based on the model performance classification scheme designed by Moriasi et al. (2007), Gauge and CHIRPS achieved "very good" performance for both the calibration and validation periods, …"

P15-L336 – "CPC showed significant overestimation in 2017 and 2018 during the verification period." was corrected as "CPC showed significant overestimation in 2017 and 2018 during the validation period."

6. L451- "although they performed slightly better at the daily scale." the model should perform better at the monthly scale?

**Authors' Response:** We apologize for this error, and we have revised it as follows:

[revised manuscript text omitted]